# Efficient Fine-Tuning of Quantized LLMs via Three-Stage Optimization

## Abstract

Fine-tuning large language models (LLMs) is computationally expensive and memory-intensive due to their vast number of parameters. To mitigate these challenges, Parameter-Efficient Fine-Tuning (PEFT) methods and model quantization techniques have been developed. Recent works have combined PEFT with quantization, proposing methods to adjust quantized model parameters before fine-tuning to reduce quantization errors. However, we observe that such adjustments can lead to suboptimal performance, as they may introduce discrepancies between the quantized and original models. Additionally, the inherent fragility of quantized models makes them sensitive to increased training complexity, potentially degrading performance. To address these issues, we introduce **QR-Adaptor**, a general fine-tuning framework that jointly optimizes quantization bit-widths and LoRA ranks for each layer in a gradient-free manner. Our method directly uses actual performance and memory usage as optimization objectives, bypassing network errors introduced by quantization. Through a three-stage optimization process—initialization based on task-specific layer importance, global exploration using a Pareto ranking genetic algorithm, and local refinement with Bayesian optimization—QR-Adaptor efficiently identifies optimal configurations. Experimental results demonstrate that QR-Adaptor yields fine-tuned low-bit quantized models that outperform their 16-bit counterparts while maintaining similar memory usage to 4-bit models. For instance, on the MMLU benchmark, our method achieves a 3.3% accuracy improvement over methods like LoftQ and LQ-LoRA.

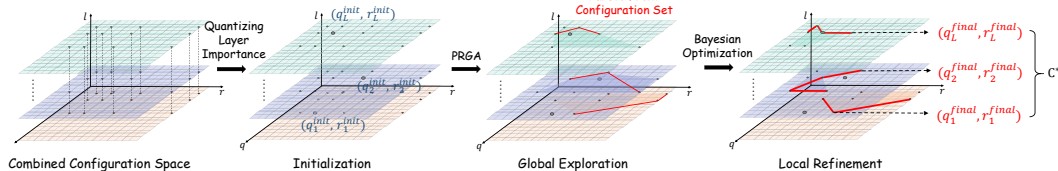

**Figure 1:** Overview of the QR-Adaptor framework: For each LLM layer, the optimal quantization bits (q) and LoRA rank (r) are determined through three steps: (1) task-based initialization, (2) PRGA global search for Pareto frontier solutions, and (3) Bayesian optimization for local refinement. The sub-graphs show: (a) the full configuration space across layers l, (b) an initial solution, (c) PRGA-identified Pareto fronts per layer, and (d) final Bayesian-optimized solutions meeting specific performance-memory trade-offs.

## 1 Introduction

Large Language Models (LLMs) have achieved unprecedented success across various natural language processing tasks (Makridakis et al., 2023; Raiaan et al., 2024; Chang et al., 2024), demonstrating exceptional capabilities in both language understanding and generation. However, adapting these models to specific downstream tasks remains challenging due to significant computational and memory constraints (Wan et al.). To address these issues, Parameter-Efficient Fine-Tuning (PEFT) methods, such as Low-Rank Adaptation (LoRA) (Hu et al., 2022), have emerged, introducing low-rank matrices to approximate updates to pre-trained weights, thereby enabling efficient fine-tuning. Meanwhile, model quantization techniques (Gong et al., 2014; Gupta et al., 2015) reduce weight precision to decrease computational costs, enhancing training and inference efficiency.

Recent advancements, such as QLoRA (Dettmers et al., 2023), integrate PEFT with quantization techniques to improve fine-tuning efficiency and achieve higher-performing quantized models. LoftQ (Li et al., 2023) and LQ-LoRA (Guo et al., 2024) propose minimizing the Frobenius norm $\|\mathbf{W} - \mathbf{Q} - \mathbf{AB}\|_F$ by adjusting the parameters of quantized weight matrix $\mathbf{Q}$, low-rank matrices $\mathbf{A}$ and $\mathbf{B}$, using this initialization to reduce the error between quantized models and full-precision models. However, this initialization only fits a small portion of the error, and the resulting model $\mathbf{Q}' + \mathbf{A}'\mathbf{B}'$ no longer equals the original pre-trained model weight matrix $\mathbf{W}$ or the quantized model weight matrix $\mathbf{Q}$ (i.e., $\mathbf{Q}' + \mathbf{A}'\mathbf{B}' \neq \mathbf{W}$ and $\neq \mathbf{Q}$). Fine-tuning this model does not necessarily lead to better performance, and in some cases, it may even perform worse than directly using the quantized model. In addition, another way to save resources is to reduce the number of parameters in the low-rank matrix. For example, AdaLoRA (Zhang et al., 2023b) dynamically prunes the rank of the low-rank matrix during fine-tuning based on its importance score. This dynamic pruning process introduces additional complexity and may lead to unexpected performance issues, as continuously adjusting the rank forces the model to adapt to a constantly changing parameter space. Such dynamic adjustments during fine-tuning are not well-suited for quantized models. In quantized models, the errors introduced by quantization already degrade the model's robustness (Gong et al., 2024), significantly weakening its ability to capture meaningful features.

We validate these two hypotheses in Section 2: $\mathbf{Q}' + \mathbf{A}'\mathbf{B}' \neq \mathbf{Q}$ before fine-tuning may lead to performance degradation, and changing the trainable parameters during fine-tuning may also cause performance issues. Based on these observations, we propose two strict constraints for fine-tuning quantized models: first, before fine-tuning, $\mathbf{Q}' + \mathbf{A}'\mathbf{B}' = \mathbf{Q}$, ensuring consistency at the starting point of fine-tuning; second, keeping the number of trainable parameters unchanged during fine-tuning to ensure that the quantized model effectively captures essential features.

Under these two constraints, and beyond approaches like QLoRA(Dettmers et al., 2024), achieving efficient fine-tuning of quantized models necessitates a strategic allocation of limited computational and memory resources to maximize performance. Specifically, we propose assigning different quantization bit-widths and LoRA ranks to various layers of the model, not solely based on their importance to the downstream task but also considering each layer's adaptability and expressiveness after quantization. By allocating higher precision (i.e., larger bit-widths) and larger LoRA ranks to layers that require more capacity to adapt to the task—thereby granting them additional computational resources—and assigning lower precision and smaller ranks to layers that maintain sufficient expressiveness even under quantization, we enhance the model's performance where it is most needed without excessively increasing memory usage.

Furthermore, to avoid introducing additional approximation errors, we employ actual task performance and memory consumption as indicators to guide the allocation process. However, determining the optimal assignment of bit-widths and ranks across layers results in a vast combinatorial solution space, and real-world evaluations are computationally intensive and time-consuming. Traditional methods, such as exhaustive enumeration or linear programming, become impractical in this context due to their high computational cost.

To tackle these challenges, we reformulate the problem as a gradient-free optimization task and introduce a three-stage optimization framework, **QR-Adaptor**, which efficiently navigates the solution space through initialization, interpolation, and extrapolation (see Figure 1). Our approach comprises: **Task-Informed Initialization**, where we derive initial layer configurations based on each layer's adaptability and contribution to the task; **Global Exploration with Pareto Ranking Genetic Algorithm (PRGA)**, inspired by NSGA-II (Deb et al., 2002), to effectively explore the broad configuration space and identify optimal trade-offs between performance and memory usage; and **Local Refinement with Bayesian Optimization**, where we employ customized weighted objective functions to refine configurations, constructing surrogate models that approximate the performance landscape and selecting optimal fine-tuned configurations. To accelerate the search process, we utilize a subset of the dataset for fine-tuning during optimization. Experimental results demonstrate that the low-precision models fine-tuned with QR-Adaptor outperform the 16-bit fine-tuned models, while maintaining memory usage comparable to that of 4-bit quantized models during fine-tuning and without requiring any structural adjustments, thereby showcasing its generalizability and effectiveness.

## 2 DISCUSSION ON FINE-TUNING QUANTIZED MODELS

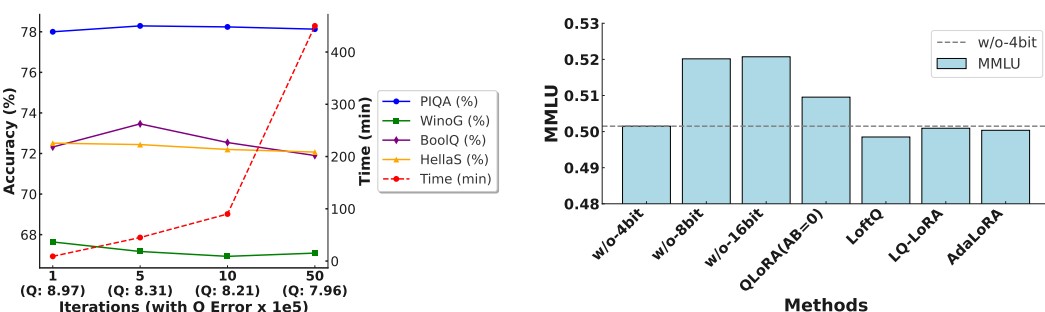

**Figure 2:** Left: After 50 iterations, it takes around 450 minutes. Compared to iter=1, the error drops from $8.97 \times 10^5$ to $7.96 \times 10^5$, but LLaMA2-7B performance shows no significant improvement. Right: Performance comparison between quantized models with/without fine-tuning (LLaMA2-13B).

LoftQ and LQ-LoRA integrate low-rank adaptation with quantization, aiming for low-precision fine-tuning of large language models. They initialize the model weights by solving the following optimization problem:

$$\min_{\mathbf{Q},\mathbf{A},\mathbf{B}} |\mathbf{W} - \mathbf{Q} - \mathbf{AB}|_F, \tag{1}$$

where $|\cdot|_F$ denotes the Frobenius norm. By alternately optimizing $\mathbf{Q}$ and $\mathbf{AB}$, they aim to reduce the error introduced by quantization. However, we find that this optimization can only capture a portion of the quantization error, and even after investing considerable time in iterations, the reduction in error is minimal and does not contribute to performance improvement (see Figure 2). More importantly, fine-tuning 4-bit quantized models using LoftQ or LQ-LoRA sometimes results in worse performance than the quantized model without fine-tuning (see Figure 2). In contrast, initializing with $\mathbf{AB} = 0$ can enhance the performance of the quantized model. This suggests that the initial optimization of $\mathbf{Q}$ and $\mathbf{AB}$ may introduce noise, adversely affecting learning and causing the fine-tuned performance to be even worse than the original $\mathbf{Q}$, because $\mathbf{Q}' + \mathbf{A}'\mathbf{B}' \neq \mathbf{W}$ and $\neq \mathbf{Q}$.

On the other hand, adaptive methods like AdaLoRA dynamically adjust the rank values of low-rank matrices based on importance scores derived from gradient norms. While effective for full-precision models, this approach faces challenges in quantized models. First, model quantization reduces robustness, making it less sensitive in capturing features. The training process of dynamically changing trainable parameters requires the model to continuously adapt to these changes, which is relatively difficult for quantized models. Second, assuming that the optimization directions of the quantized model and the full-precision model are consistent (since high-precision models always have greater representational capacity than low-precision ones), the error introduced by quantization can distort the gradient

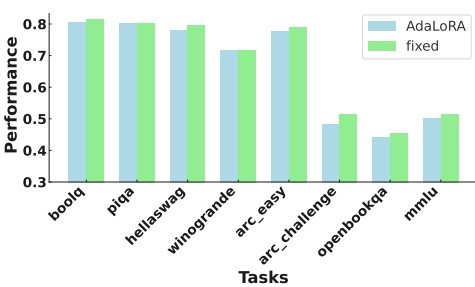

**Figure 3:** Performance comparison of two fine-tuning methods using AdaLoRA and random assignment of different rank values for each layer but fixed trainable parameters during fine-tuning.

norms, leading to unreliable importance scores. Due to error accumulation, this distortion is exacerbated in deeper layers. Dynamic rank adaptation based on flawed importance scores may result in improper resource allocation, thus hindering learning. Our experimental results also validate this point (see Figure 2). Fine-tuning quantized models using AdaLoRA does not yield satisfactory performance. In contrast, configurations with randomly assigned average ranks equal to the target rank of AdaLoRA achieve better performance after fine-tuning (see Figure 3). The key difference lies in whether the number of trainable parameters remains fixed during fine-tuning.

According to the above discussion, we propose two key constraints for effectively fine-tuning quantized models: **Preserve quantized model parameters before fine-tuning.** Before fine-tuning, ensure that the sum of quantized weights and low-rank updates equals the quantized model itself,

This constraint implies that the initial low-rank updates do not alter the parameters of the quantized model, providing a stable starting point for fine-tuning. Unlike previous methods that adjust $\mathbf{Q}$, $\mathbf{A}$, and $\mathbf{B}$ to approximate $\mathbf{W}$, we keep $\mathbf{Q}$ unchanged to avoid introducing additional discrepancies. **Keep the number of trainable parameters fixed to reduce training difficulty.** To reduce training complexity and ensure that quantized models can capture meaningful features for downstream tasks, we keep the number of trainable parameters fixed during fine-tuning, eliminating the need for the quantized model to adapt to a constantly changing parameter space. This constraint is in contrast to the traditional approach for high-precision models, where increasing training complexity is necessary. Since quantized models have lower representational capacity, increasing training difficulty is not a wise choice.

## 3 METHODOLOGY

We begin this section by framing the problem as a gradient-free optimization challenge. Once the necessary background has been introduced, we then propose a novel three-stage optimization algorithm specifically designed to tackle this complex task.

### 3.1 PROBLEM FORMULATION

Given a pre-trained LLM with $L$ layers, our objective is to fine-tune the model on a training dataset $\mathcal{D}_{\text{train}}$ while both maximizing its performance on downstream evaluation datasets $\mathcal{D}_{\text{test}}$ and minimizing the model's memory footprint.

**Layer-wise LoRA**   In LoRA fine-tuning, the forward pass of a layer incorporates a low-rank adaptation so that:

$$\mathbf{y} = \mathbf{W}_l \mathbf{x} + \Delta \mathbf{W}_l \mathbf{x}, \tag{2}$$

where $\mathbf{x} \in \mathbb{R}^k$ is the input vector, $\mathbf{W}_l \in \mathbb{R}^{d \times k}$ is the weight matrix for layer $l \in \{1, \cdots, L\}$, and $\Delta \mathbf{W}_l = \mathbf{A}_l \mathbf{B}_l$ represents the low-rank adaptation. The matrices $\mathbf{A}_l \in \mathbb{R}^{d \times r_l}$ and $\mathbf{B}_l \in \mathbb{R}^{r_l \times k}$ are low-rank matrices with rank $r_l$, where $r_l \in \mathcal{R}$, and $\mathcal{R}$ represents the set of all possible rank values.

**Layer-wise Quantization**   On the other hand, the quantized weight matrix $\hat{\mathbf{W}}_l$ is obtained by applying a quantization function to the weight matrix $\mathbf{W}_l$:

$$\hat{\mathbf{W}}_l = \text{Quantize}(\mathbf{W}_l, q_l), \tag{3}$$

where $q_l$ denotes the bit-width used for quantization in layer $l$, with $q_l \in \mathcal{Q}$, and $\mathcal{Q}$ represents the set of all possible bit-width values.

**Integrating Layer-wise LoRA and Quantization**   When quantization is combined with LoRA, we first quantize the weight matrix and then implement LoRA fine-tuning:

$$\mathbf{y} = \hat{\mathbf{W}}_l^{q_l} \mathbf{x} + \Delta \mathbf{W}_l^{r_l} \mathbf{x}, \tag{4}$$

where $\hat{\mathbf{W}}_l^{q_l}$ represents the weight matrix quantized with $q_l$ bits, and $\Delta \mathbf{W}_l^{r_l}$ can be decomposed into two $r_l$-rank matrices.

**Weighted Objective Function**   Finding the optimal $q_l$ and $r_l$ for each layer can be formulated as a gradient-free optimization problem. Let $C = \{(q_1, r_1), (q_2, r_2), \ldots, (q_L, r_L), q_l \in \mathcal{Q}, r_l \in \mathcal{R}\} \in \mathcal{C}$ represent the fine-tuning configuration of an $L$-layer LLM, where $\mathcal{C}$ is the configuration space consisting of all combinations of bit-width and rank values. Our objective is to find an optimal configuration set $C^*$ for efficient fine-tuning that achieves the best performance on downstream tasks while minimizing memory usage. To balance these competing goals, we introduce a weighted objective function. Put formally, this process can be formulated as:

$$\max_C \quad \alpha \cdot \frac{P(C) - \mu_P}{\sigma_P} - (1 - \alpha) \cdot \frac{M(C) - \mu_M}{\sigma_M},$$
$$\text{subject to} \quad C = \{(q_l, r_l)_{l=1}^L\} \in \mathcal{C}, \quad q_l \in \mathcal{Q}, \quad r_l \in \mathcal{R}. \tag{5}$$

In the above, $P(\cdot)$ denotes the model's performance, and $M(\cdot)$ calculates the total memory consumption based on the configurations. The terms $\mu_P$ and $\sigma_P$ represent the mean and standard deviation of the performance metric across configurations, while $\mu_M$ and $\sigma_M$ represent the same for memory consumption. As usual, the respective $\mu$ and $\sigma$ terms normalize the performance and memory metrics to a comparable scale. The weight parameter $\alpha \in [0, 1]$ allows us to balance the relative importance of performance versus memory efficiency: a higher $\alpha$ value prioritizes performance, while a lower value emphasizes memory efficiency.

## 3.2 QR-ADAPTOR FRAMEWORK

Our approach differs from previous methods relying on fixed or hierarchical dynamic single configurations. We jointly explore the configuration space of quantization bits and LoRA ranks, creating a comprehensive search space that encompasses all potential optimal configurations. The main challenges in implementing this gradient-free optimization process are (a) The high-dimensional, discrete nature of the configuration space. (b) The computational cost of evaluating performance. To address these challenges, we propose QR-Adaptor, a method that effectively finds the relative optimal solution in three stages.

**Task Information Based Initialization** Our optimization process begins with a task-oriented assessment of the relative importance of each layer in the model. This approach is based on the tacit understanding that different layers contribute unequally to the model's overall performance for specific tasks. Unlike previous methods that relied on gradient norms to quantify layer importance—an approach that fails to accurately represent a layer's contribution during inference—we employ a task-specific method based on information entropy during the inference process. We define the importance of a layer $l$ for a given task as:

$$I(l) = H(Y) - H(Y|X_l) \tag{6}$$

In the above, $H(Y)$ is the entropy of the model's output for the task, and $H(Y|X_l)$ is the conditional entropy of the output given the intermediate representation at layer $l$. This measure quantifies how much information each layer contributes to the final output, providing a more accurate representation of layer importance in the context of the specific task. This task-oriented approach allows us to strategically allocate higher bit widths and ranks to layers that are critical for the given task, rather than relying on generic importance metrics that may not reflect true inference contributions.

We initialize the per-layer quantization configurations using the importance scores derived from the original model. Specifically, leveraging our guiding metric, we assign higher quantization bit numbers to layers with higher importance scores, while allocating lower values to less critical layers; once assigned, we quantize the model according to these assigned bit widths. Following quantization, we recalculate the importance scores and use them to determine the LoRA rank values. Layers with higher post-quantization importance scores are assigned larger rank values, while those with lower scores receive smaller ones. This informed initialization reduces the search space and guides the optimization towards promising regions.

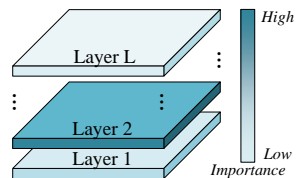

**Figure 4:** Different Layers have heterogeneous importance

**Global Exploration with PRGA** In LLM fine-tuning, quantization bit and LoRA rank can be conceptualized as genes, with the resulting performance and memory usage analogous to phenotypic expressions in a population. Inspired by NSGA-II's (Deb et al., 2002) proven success in multi-objective optimization, we adapted its mechanisms to develop PRGA (Pareto Ranking Genetic Algorithm), incorporating domain-specific modifications to better handle the discrete-continuous hybrid search space of LLM fine-tuning hyperparameters. PRGA explores the combined solution space of bits and ranks to identify the optimal Pareto frontier, simultaneously balancing performance and memory usage in LLM fine-tuning. This efficient multi-objective optimization algorithm uses an elitist selection approach to evolve a population of configurations, each represented as $C = \{(q_l, r_l)_{l=1}^L\}$, where $q_l$ and $r_l$ denote the quantization bits and LoRA rank for layer $l$, respectively. PRGA iteratively applies selection, crossover, and mutation operations to the population, aiming to simultaneously maximize performance and minimize memory usage. The algorithm progresses until it reaches a predefined stopping criterion or a maximum number of iterations,

whichever comes first. Once finished, the algorithm ultimately produces a Pareto frontier that represents the optimal trade-offs between our competing objectives.

**Algorithm 1** Pareto Rank Calculation

1: Calculate the number of dominated individuals $n_p$ and the set of solutions dominated $S_p$ for each individual $p$
2: Place individual with $n_p = 0$ into set $F_1$
3: **for** each individual in $F_1$ **do**
4:     **for** each individual $j \in S_i$ **do**
5:         $n_j = n_j - 1$
6:         **if** $n_j = 0$ **then**
7:             Add individual $j$ to set $F_2$
8:         **end if**
9:     **end for**
10: **end for**
11: Repeat step 3 for set $F_2$ to obtain $F_3$, and continue until all individuals are ranked
12: **return** All individuals with Pareto rank

**Algorithm 2** Crowding Distance Calculation (Ranking individuals with the same Paret Rank)

1: **for** each individual $n \in 1 \ldots N$ **do**
2:     Initialize $d_n = 0$
3: **end for**
4: **for** each objective function $f_m$ **do**
5:     Sort individuals based on $f_m$
6:     Set $f_m^{max}$ and $f_m^{min}$
7:     Set $d_1 = d_N = \infty$
8:     **for** $n = 2$ to $N - 1$ **do**
9:         $d_n = d_n + \frac{f_m(n+1) - f_m(n-1)}{f_m^{max} - f_m^{min}}$
10:     **end for**
11: **end for**
12: **return** crowding distances $d_n$ for each individual $n \in 1 \ldots N$

Before introducing the PRGA flow, we first introduce some key foundational concepts. In a multi-objective minimization problem with $n$ objective components $f_i(x), i = 1, \ldots, n$, the Pareto Dominance Relationship is defined between any two decision variables $X_a$ and $X_b$. We say that $X_a$ dominates $X_b$ if for all $i \in \{1, 2, \ldots, n\}$, $f_i(X_a) \leq f_i(X_b)$, and there exists at least one $i \in \{1, 2, \ldots, n\}$ such that $f_i(X_a) < f_i(X_b)$. A Non-dominated Solution is a decision variable that is not dominated by any other decision variable in the set. The concept of Pareto Rank is used to categorize solutions within a set. Non-dominated solutions are assigned a Pareto rank of 1. After removing these rank 1 solutions from the set, the remaining non-dominated solutions are assigned a Pareto rank of 2. This process continues iteratively, assigning increasing ranks to subsequent layers of non-dominated solutions until all solutions in the set have been ranked.

As illustrated in Algorithm 1, we employ the Pareto Ranking to sort all individuals within the population. To address solutions with identical Pareto ranks, we use the crowding distance $d$ for further differentiation within each Pareto rank. The detailed calculation of the crowding distance is presented in Algorithm 2.

Elite retention, a method simulating natural elimination, is performed after calculating the Pareto rank and crowding distance of all individuals in a generation. This process begins by combining the parent and offspring populations into a merged population. To generate the next generation, we start with the lowest Pareto rank and transfer entire layers of individuals from the merged population to the new population, moving progressively to higher ranks. This continues until we reach a layer that cannot be fully accommodated in the new population. For this partially accommodated layer, we sort its individuals based on their crowding distance in descending order and add them sequentially to the new population until it reaches its full capacity.

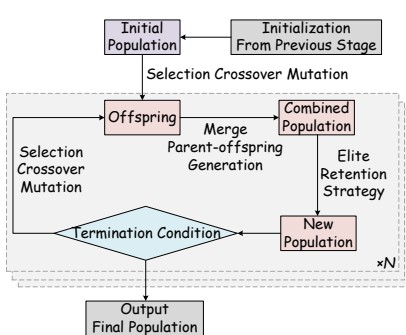

**Figure 5:** Detailed PRGA flow chart. The input is a set of solutions from the initialization, and the output is a set of Pareto front solutions containing multiple solutions.

For the crossover and mutation operations, we employ methods analogous to Simulated Binary Crossover (SBX) and Polynomial Mutation, respectively. These methods are adapted to operate on $L$ pairs of positive integers $(q_l, r_l)$. The detailed procedures for these operations are presented in Algorithm 3 for the crossover and Algorithm 4 for the mutation. By applying these adapted SBX and polynomial mutation operations, we can effectively evolve the population of solutions represented by integer pairs, balancing exploration and exploitation.

The PRGA process, shown in Figure 5, begins by generating an initial population of size $N$ through controlled random variations based on the previous stage's configuration. It then creates offspring using selection, crossover, and mutation operations. The parent and offspring populations are com-

**Algorithm 3** Simulated Binary Crossover

**Require:** Two parent individuals $P_1$ and $P_2$, each containing $L$ pairs of real numbers
1: **for** $l = 1$ to $L$ **do**
2:     Generate a random number $u \in [0, 1]$
3:     **if** $u \le 0.5$ **then**
4:         $\beta = (2u)^{1/(n+1)}$
5:     **else**
6:         $\beta = (1/(2(1-u)))^{1/(n+1)}$
7:     **end if**
8:     $y_{1l} = 0.5 \cdot ((1+\beta) \cdot p_{1l} + (1-\beta) \cdot p_{2l})$
9:     $y_{2l} = 0.5 \cdot ((1-\beta) \cdot p_{1l} + (1+\beta) \cdot p_{2l})$
10:    Add $(y_{1l}, y_{2l})$ to $O_1$ and $O_2$
11: **end for**
12: **return** Two offspring $O_1$ and $O_2$

**Algorithm 4** Polynomial Mutation

**Require:** Individual $P$ containing $L$ pairs of real numbers, mutation probability $p_m$
1: **for** $l = 1$ to $L$ **do**
2:     **for** each value $x$ in the $l$-th pair **do**
3:         Generate a number $u \in [0, 1]$
4:         **if** $u < p_m$ **then**
5:            Generate a number $y \in [-1, 1]$
6:            $x' = x + (x_{max} - x_{min}) \cdot (y \cdot (1 - |y|)^{n-1})$
7:            Replace $x$ with $x'$ in $P'$
8:         **end if**
9:     **end for**
10: **end for**
11: **return** Mutated individual $P'$

bined and subsequently fastly fine-tuned and validated on a subset of the training dataset. The resulting performance and memory metrics are used to calculate Pareto ranks for each individual. Next, the algorithm applies an elite retention strategy combined with crowding distance calculation to select individuals for the new population. This cycle repeats, generating consecutive generations until the termination condition is met, effectively exploring the solution space to optimize both performance and memory usage simultaneously.

**Local Refinement with Bayesian Optimization** While PRGA effectively explores the global configuration space, it may not precisely capture local optima near the Pareto front. To further refine these solutions, we employ Bayesian optimization, a technique renowned for its ability to optimize expensive black-box functions with uncertainty quantification. We initiate this process by utilizing the solutions from the PRGA-generated Pareto front as our initial sampling points. For each point, we use these configurations to quickly fine-tune the model and test to obtain actual performance and memory usage. We then compute its corresponding weighted objective function value $y$ using the predefined objective function (Equation 5) and specified weight preferences. These $y$ values serve a dual purpose: firstly, they are combined with the covariance matrix $K$, which is constructed using the radial basis function (RBF) kernel to quantify similarities between sample points, to build a Gaussian process model; secondly, they enable us to identify the best-performing point, which becomes the focal point for subsequent searches.

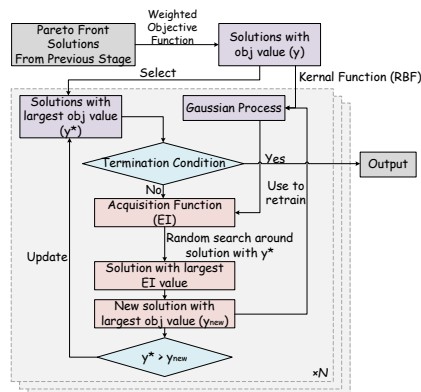

**Figure 6:** Detailed Bayesian optimization flow chart. The input is the Pareto front solution set from the global search, and the output is a set of optimal solutions obtained according to the requirements.

The next phase involves employing a random search strategy to select new sampling points in the vicinity of this top-performing configuration. For each newly selected sampling point $x^*$, we leverage the Gaussian process to estimate its predicted value and associated uncertainty using the following equations:

$$\mu(x^*) = m(x^*) + K(x^*, X)K(X, X)^{-1}(y - m(X))$$
$$\sigma^2(x^*) = k(x^*, x^*) - K(x^*, X)K(X, X)^{-1}K(X, x^*) , \tag{7}$$

where $m(x^*)$ is the prior mean function, $K(x^*, X)$ is the covariance between the new point and the existing points, and $K(X, X)$ is the covariance matrix of the existing points.

We then employ the Expected Improvement (EI) as the acquisition function, calculating it using the following formula:

$$EI(x^*) = \sigma(x^*) \left( Z \cdot \Phi(Z) + \phi(Z) \right) \quad \text{and} \quad Z = \frac{\mu(x^*) - y_{\text{best}}}{\sigma(x^*)} \tag{8}$$

where $\Phi(Z)$ is the cumulative distribution function of the standard normal distribution, while $\phi(Z)$ is its probability density function and $y_{best}$ is the best objective value among all points.

We select the point with the largest EI value as the next evaluation point and quickly fine-tune the LLM using this configuration. We then test to obtain the performance and memory usage, which are used to calculate the objective function value, and this new point is compared with the previous best point. If it proves to be a better choice, it can be updated as the optimal point and the search for the next iteration proceeds near it; otherwise, we continue searching near the original optimal point. Regardless of whether the best point is updated, the new point and its objective function value are incorporated into the training dataset to update the Gaussian process model before beginning the next iteration. This process repeats until a predefined termination condition is met, such as reaching maximum iterations, meeting a convergence criterion, or hitting a time limit. The detailed flowchart is shown in Figure 6.

The solution set obtained through this Bayesian optimization process offers a refined representation of high-quality configurations, effectively capturing the trade-offs between performance and memory usage. This iterative refinement process culminates in a final set of configurations that represent the best balance of our objectives based on the specified preferences. By presenting these optimized configurations as the final output, our approach enables practitioners to directly choose suitable configuration that aligns with their specific performance requirements and memory constraints.

# 4 EVALUATION

**Table 1:** Superscripts on LoftQ bits indicate the number of initialization iterations. QR-Adaptor searches for optimal bit-width and rank value for each layer based on different tasks; its bit number and peak memory usage are averaged across 7 tasks. Bold figures represent the best performance for a given model and task, while underlined indicate the second-best. Accuracy is reported as %, and memory is measured in GB.

| | Method | Bit | BoolQ | PIQA | HellaS | WinoG | ARC-e | ARC-c | OBQA | Average | Memory |
|---|---|---|---|---|---|---|---|---|---|---|---|
| Llama 2-13B | w/o tuning | 16 | 80.61 | 80.52 | 79.37 | 72.06 | 79.46 | 49.15 | 45.20 | 69.48 | - |
| | | 8 | 79.94 | 80.20 | 79.14 | 72.61 | 78.91 | 48.89 | 45.40 | 69.30 | - |
| | | 4 | 80.52 | 79.98 | 78.38 | 71.59 | 77.65 | 48.29 | 44.80 | 68.74 | - |
| | LoRA | 16 | 81.50 | 81.23 | 80.07 | 71.98 | 79.84 | 52.13 | **46.20** | 70.42 | 41.13 |
| | QLoRA | 8 | 81.13 | 81.18 | 79.86 | 72.22 | 80.01 | 51.54 | **46.20** | 70.31 | 38.28 |
| | | 4 | 81.04 | 80.47 | 79.48 | 71.82 | 79.04 | 51.45 | 45.60 | 69.84 | 27.30 |
| | AdaLoRA | 16 | 80.46 | 80.47 | 79.28 | 72.30 | 79.34 | 49.40 | 45.40 | 69.52 | 41.08 |
| | | 8 | 80.40 | 80.52 | 79.27 | 72.38 | 79.29 | 49.49 | 45.40 | 69.54 | 38.24 |
| | | 4 | 80.43 | 80.09 | 78.10 | 71.67 | 77.69 | 48.29 | 44.20 | 68.64 | 27.30 |
| | LoftQ | $4^1$ | 80.86 | 80.30 | 79.18 | 71.90 | 78.87 | 50.68 | 45.80 | 69.66 | 41.02 |
| | | $4^5$ | 80.92 | 80.41 | 79.15 | 71.59 | 78.96 | 50.60 | 45.40 | 69.58 | 41.03 |
| | LQ-LoRA | 4 | 80.43 | 80.14 | 79.06 | 71.67 | 78.79 | 50.09 | 45.40 | 69.37 | 39.65 |
| | QR-Adaptor | 6.125 | **81.84** | **81.45** | **80.08** | **72.69** | **80.64** | **52.82** | 45.80 | **70.76** | 27.41 |
| Llama 2-7B | w/o tuning | 16 | 77.68 | 79.11 | 76.01 | 68.98 | 76.30 | 46.16 | 44.20 | 66.92 | - |
| | | 8 | 77.58 | 79.27 | 76.04 | 68.98 | 75.97 | 46.50 | 44.00 | 66.91 | - |
| | | 4 | 76.21 | 78.18 | 75.57 | 69.06 | 75.25 | 45.99 | 44.40 | 66.38 | - |
| | LoRA | 16 | 78.41 | 79.38 | 76.81 | 69.06 | **77.57** | 46.93 | 45.00 | 67.59 | 23.61 |
| | QLoRA | 8 | 78.41 | 79.05 | **76.93** | 69.06 | 77.44 | 47.61 | 45.40 | 67.70 | 23.51 |
| | | 4 | 77.25 | 78.84 | 76.40 | 70.01 | 76.35 | 46.67 | 45.00 | 67.22 | 17.53 |
| | AdaLoRA | 16 | 77.58 | 79.11 | 75.92 | 69.38 | 76.68 | 46.16 | 44.20 | 67.00 | 23.56 |
| | | 8 | 77.40 | 79.11 | 75.91 | 69.06 | 76.68 | 46.16 | 44.40 | 66.96 | 23.49 |
| | | 4 | 76.45 | 77.91 | 75.44 | 69.46 | 75.29 | 46.33 | 44.20 | 66.44 | 17.26 |
| | LoftQ | $4^1$ | 77.89 | 79.43 | 76.61 | 69.69 | 77.19 | 47.10 | 44.80 | 67.53 | 23.75 |
| | | $4^5$ | 76.79 | 78.51 | 76.25 | 69.61 | 76.47 | 47.95 | 45.60 | 67.31 | 23.82 |
| | LQ-LoRA | 4 | 77.22 | 78.78 | 76.33 | 70.09 | 76.39 | 47.10 | **46.40** | 67.47 | 22.84 |
| | QR-Adaptor | 5.875 | **78.96** | **79.86** | 76.84 | 69.97 | 77.44 | **48.04** | 46.00 | **68.15** | 17.92 |

We conduct experiments to evaluate our proposed method against various baselines. All hyperparameters aside from rank value and bit-width are kept consistent with the baselines. Additionally, we performed an ablation study to assess the impact of each stage on performance.

**Datasets and LLMs.** We utilize the Alpaca52k and hc3 (Taori et al., 2023) [1] for fine-tuning and evaluate the zero-shot performance of these LLMs on benchmarks including BoolQ (Clark et al., 2019), PIQA (Bisk et al., 2020), HellaSwag (Zellers et al., 2019), WinoGrande (Sakaguchi et al., 2021), ARC-easy (Clark et al., 2018), ARC-challenge (Clark et al., 2018), OpenbookQA (Mihaylov et al., 2018), and MMLU (Hendrycks et al., 2021). The models used in our experiments are LLaMA2 (Touvron et al., 2023) and LLaMA3.1 (Grattafiori et al., 2024).

**Baselines.** We compare our method against several baselines: without tuning, LoRA (Hu et al., 2022), QLoRA (Dettmers et al., 2023), Adalora (Zhang et al., 2023b), LoftQ (Li et al., 2023), and LQ-LoRA (Guo et al., 2024). We evaluated the performance of LoftQ with different iteration numbers. For Adalora, which dynamically allocates ranks based on the average rank budget, we set the budget to 8 and 64. Finally, for LQ-LoRA, which allocates quantization bit-width based on the average weight bit-width budget and quantization error, we set the bit-width budget to 4.

**Implementation Details.** We utilize the following configurations: *PyTorch* version 2.1.2, *Bitsand-Bytes* library version 0.43.1, *Transformers* library version 4.41.0, *PEFT (Parameter-Efficient Fine-Tuning)* library version 0.11.1, *Optuna* library version 3.6.1, *CUDA* version 12.4, *GPU:* NVIDIA L20 GPU. *Operating System:* Ubuntu. Concise implementation details are provided in the appendix D. In our framework, we define the population size as 5 and generate 1 new offspring in each iteration. The second stage runs for 5 iterations, and similarly, the third stage also iterates 5 times.

## 4.1 MAIN RESULTS

We present the performance comparison on commonsense understanding tasks in Table 1, with more results in the appendix B. The results for the MMLU task in LLaMA2 are shown in Figure 7. QR-Adaptor demonstrates outstanding performance across various benchmarks. Due to the rank value selection ranging from 2 to 16, in some cases, QR-Adaptor consumes less memory than the fine-tuned 4-bit quantized models. Moreover, the low-precision models fine-tuned by QR-Adaptor outperform the fine-tuned 16-bit models. Another advantage of the QR-Adaptor is that it can be implemented without any additional technical measures to optimize performance, apart from spending some time (about 15 minutes to get one data point). This simple but effective method is very useful in practical applications.

Due to hardware constraints, we did not test models larger than 70B, but compared to other methods, QR-Adaptor can iteratively optimize larger models on the same hardware. Existing research shows that modifying only a subset of parameters can significantly change performance, which implies that applying our method to larger-scale models would not greatly increase time consumption, as iteration optimization can be achieved by reducing fine-tuning data and conducting rapid evaluations.

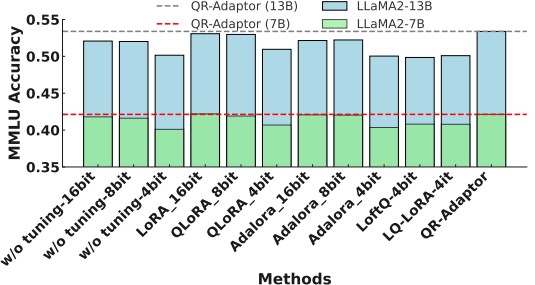

**Figure 7:** Performance comparison on MMLU benchmark. QR-Adaptor outperforms other methods.

Additionally, the experimental results indicate that the two problems we discussed earlier regarding fine-tuning quantized models persist, especially with the 13B model. Despite our efforts to select appropriate configurations for the baseline methods, their performance is still inferior to the simplest QLoRA. For the MMLU task, baseline methods may perform even worse than quantized models without tuning.

## 4.2 ABLATION STUDY

We use the WinoGrande benchmark as an example for the ablation study to evaluate the role of each stage in QR-Adapto. As shown in Figure 8, it is evident that excluding PRGA and Bayesian optimization leads to uneven exploration of the search space—one is too broad and the other too concentrated—since they represent the extrapolation and interpolation capabilities, respectively. Excluding stage 1 results in overly scattered exploration because PRGA starts from a random search

---
[1] https://huggingface.co/datasets/yahma/alpaca-cleaned

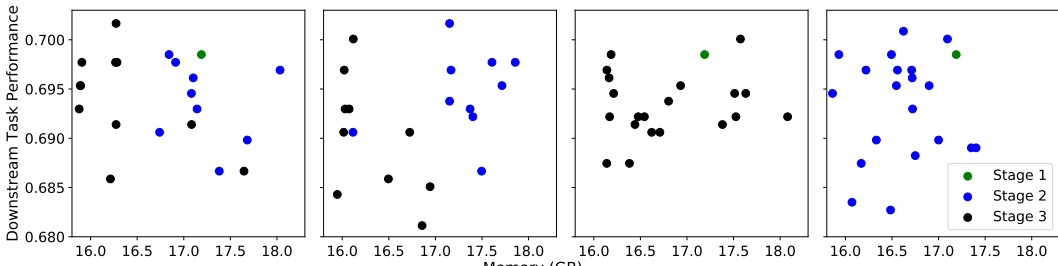

**Figure 8:** From left to right, the actual measured performance and memory usage of the configurations generated by QR-Adaptor, QR-Adaptor without stage1, QR-Adaptor without stage2, and QR-Adaptor without stage3 are shown. Different colors represent the configurations generated at different stages.

without an initialization point. However, it still manages to explore the theoretically optimal region in the upper-left corner, demonstrating the strong capabilities of PRGA and Bayesian optimization. In contrast, the complete three-stage QR-Adaptor clearly shows the advantage of first conducting a broad exploration around the initialization point, followed by interpolation near promising solutions to further optimize and identify the best configuration. Other ablation in the appendix E.

## 5 RELATED WORK

**LLM Quantization.** The field of LLM quantization has witnessed substantial progress, driven by the need for efficient model deployment. Recent research has introduced several innovative approaches. Frantar et al. (2023) have developed GPTQ, which achieves 4-bit precision with layer-wise quantization. Lin et al. (2023) have proposed AWQ, which improves accuracy for heavily quantized models. Yao et al. (2022) have introduced ZeroQuant, which preserves zero-shot capabilities at lower bit widths. Dettmers et al. (2022) have presented LLM.int8(), which enables 8-bit quantization for consumer hardware. Kim et al. (2023) have combined quantization with pruning and knowledge distillation in SqueezeLLM. Guan et al. (2024) have optimized the balance between compression and performance through mixed-precision quantization with APTQ. These developments significantly enhance the efficiency and accessibility of large language models.

**Parameter Efficient Fine-Tuning.** PEFT techniques have become crucial for enhancing LLMs without increasing inference overhead. Recent innovations have expanded the field. Dettmers et al. (2023) have introduced QLoRA, which combines 4-bit quantization with low-rank adapters. Li et al. (2023) have presented LoftQ, which alternates between quantization and low-rank approximation steps. Berman & Peherstorfer (2024) have introduced CoLoRA for accelerating the prediction of solution fields under new parameters. AdaLoRA (Zhang et al., 2023a) proposes adaptive budget allocation for low-rank updates, while LQ-LoRA (Guo et al., 2023) combines low-rank decomposition with quantization for efficient fine-tuning under memory constraints. Additionally, Zhou et al. (2024) have introduced RankAdaptor, which is a hierarchical dynamic low-rank adaptation method for structural pruned LLMs. These advancements demonstrate the evolving landscape of PEFT techniques, offering innovative solutions for efficient LLM fine-tuning across diverse applications.

## 6 CONCLUSION

We have identified the issues arising in the current fine-tuning of quantized models and have established two constraints accordingly. Under these constraints, the performance of fine-tuning quantized models will at least not be worse than before fine-tuning. To achieve higher performance in low-bit models while saving memory during fine-tuning, we propose QR-Adaptor, a general and efficient fine-tuning framework. It enables low-bit models to outperform fine-tuned models at the original precision. Based on our experimental results, we found that altering the bit-width of each layer and adjusting the allocation of trainable parameters can lead to significant shifts in performance, and this trend is largely predictable by the algorithm. In theory, our framework is also applicable to high-precision models, but this paper primarily focuses on fine-tuning under quantization.

REPRODUCIBILITY STATEMENT

To ensure the reproducibility of our results, we provide comprehensive documentation on the steps required to replicate our experiments. Our code is available in scripts such as `optuna_main-v3.py`, `post_training_mixed_quant.py`, and `run_optuna.py`, which handle hyperparameter optimization, mixed-precision quantization, and evaluation. For data preparation, we utilize the Alpaca Cleaned Dataset from `yahma/alpaca-cleaned`, which is automatically downloaded and processed using the `datasets` library. Our environment setup requires an NVIDIA GPU with CUDA support, preferably with at least 20 GB of memory for the LLaMA 2 model, as well as Python 3.8+ and dependencies like PyTorch, Transformers, Optuna, BitsAndBytes, PEFT, and other libraries, which can be installed via the `requirements.txt` file. The model we fine-tune is the LLaMA 2 architecture (`NousResearch/Llama-2-7b-hf`), using a mixed-precision quantization approach via `bitsandbytes` and Low-Rank Adaptation (LoRA) with the `peft` library. The training is conducted using a mixed-precision setup where the model's dtype is set to `torch.bfloat16` to optimize memory usage and computation efficiency. Our hyperparameter optimization framework leverages Optuna to maximize model accuracy while minimizing memory usage, tuning parameters like quantization bits (4 or 8 bits) and LoRA ranks (2 to 16). To replicate our training process, researchers can execute the provided scripts using the specified command-line arguments, which configure the model, output directories, number of trials, and evaluation tasks. Model checkpoints and Optuna results are saved at regular intervals. The training is conducted using the Hugging Face `Trainer`, configured with parameters including a batch size of 4, gradient accumulation steps of 16, warmup steps of 100, and a learning rate of 1e-4, with evaluation and model saving steps set to every 200 steps. Evaluation is conducted using the `lm_eval` library, where metrics such as accuracy are recorded and saved in JSON format. All hyperparameter settings and model configurations are logged in the output directory, along with training progress and memory usage. Random seeds are set to ensure deterministic behavior. By following these steps, including hardware and software specifications, and running the scripts with the provided configurations, researchers can reproduce our experiments and validate the findings related to mixed-precision quantization and parameter-efficient fine-tuning.

ETHICS STATEMENT

This work builds upon pre-trained large language models LLaMA-2 and utilizes publicly available datasets for instruction fine-tuning Alpaca-clean. We do not introduce any new datasets or data collection processes, and therefore do not involve human annotation in this research. Additionally, our study focuses on improving model efficiency through pruning and quantization techniques, without engaging with sensitive content or user-specific data. As such, this paper does not present any ethical concerns beyond those already associated with the broader body of research on large language models and their datasets. All datasets and models used comply with their respective licenses and terms of use.

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

# A    QUANTIZATION

We first apply NF-quantization with bit size $b_0$ and bucket size $B_0$ to obtain the quantized matrix $\widehat{A_i}$ and the absmax values for each block $s = [s_1, \ldots, s_{\frac{\text{sizeof}(A_i)}{B_0}}]$. These absmax values are further quantized to $b_1$ bits via uniform integer quantization with bucket size $B_1$ to obtain the quantized vector $\widehat{s}$, along with the absmax values for $s$, i.e., $v = [v_1, \ldots v_{\frac{\text{sizeof}(A_i)}{B_0 B_1}}]$. Finally, we cast $v$ to $b_2$ bits to obtain $\widehat{v}$.

This quantization scheme requires storing $\widehat{A_i}, \widehat{s}, \widehat{v}$ to represent $A_i$. We can thus quantify the memory cost (number of bits) for storing $A_i$ given a configuration $c_i = (b_0, b_1, b_2, B_0, B_1)$ as:

$$\text{memory\_cost}(A_i, c_i) = \text{sizeof}(A_i) \cdot \left( b_0 + \frac{b_1}{B_0} + \frac{b_2}{B_0 \cdot B_1} \right) \tag{9}$$

The original NF-4 double quantization is a special case with $q_{\text{NF4}} = (4, 8, \texttt{fp32}, 64, 256)$ and $\text{memory\_cost}(A_i, q_{\text{NF4}}) = 4.127 \cdot \text{sizeof}(A_i)$, i.e., NF-4 requires on average 4.127 bits per parameter.

# B    MORE RESULTS

Due to page limitations, we present all the results of rank=8 and the comparison with QR-Adaptor here.

**Table 2:** Performance comparison of different methods (rank=8) across various bit-width configurations. Superscripts on LoftQ bits indicate the number of initialization iterations. QR-Adaptor searches for optimal bit number and rank value for each layer based on different tasks; its bit number and peak memory usage are averaged across 7 tasks. Bold figures represent the best performance for a given model and task, while underlined figures indicate the second-best. Accuracy is reported as %, and memory is measured in GB.

| | Method | Bit | BoolQ | PIQA | HellaS | WinoG | ARC-e | ARC-c | OBQA | Average | Memory |
|---|---|---|---|---|---|---|---|---|---|---|---|
| | w/o tuning | 16 | 80.61 | 80.52 | 79.37 | 72.06 | 79.46 | 49.15 | 45.20 | 69.48 | - |
| | | 8 | 79.94 | 80.20 | 79.14 | 72.61 | 78.91 | 48.89 | 45.40 | 69.30 | - |
| | | 4 | 80.52 | 79.98 | 78.38 | 71.59 | 77.65 | 48.29 | 44.80 | 68.74 | - |
| | LoRA | 16 | 81.44 | 81.12 | 79.98 | 71.98 | 80.18 | 52.56 | **46.40** | 70.52 | 41.04 |
| Llama2-13B | QLoRA | 8 | 81.22 | 80.47 | 79.92 | **73.09** | 80.18 | 52.39 | 45.00 | 70.32 | 37.82 |
| | | 4 | 81.41 | 80.30 | 79.46 | 71.82 | 78.91 | 51.54 | 45.40 | 69.83 | 26.84 |
| | AdaLoRA | 16 | 80.37 | 80.47 | 79.25 | 72.30 | 79.46 | 49.15 | 45.40 | 69.49 | 41.07 |
| | | 8 | 80.43 | 80.47 | 79.29 | 72.22 | 79.34 | 49.32 | 45.60 | 69.52 | 38.36 |
| | | 4 | 80.40 | 80.14 | 78.12 | 71.74 | 77.78 | 48.29 | 44.20 | 68.67 | 27.30 |
| | LoftQ | $4^1$ | 81.16 | 80.41 | 79.12 | 71.35 | 78.79 | 50.68 | 45.80 | 69.62 | 40.56 |
| | | $4^5$ | 80.24 | 80.25 | 78.81 | 70.80 | 78.87 | 50.34 | 45.20 | 69.22 | 39.81 |
| | LQ-LoRA | 4 | 80.67 | 80.14 | 78.91 | 71.11 | 78.79 | 50.60 | 45.00 | 69.32 | 39.81 |
| | QR-Adaptor | 6.125 | **81.84** | **81.45** | **80.08** | 72.69 | **80.64** | **52.82** | 45.80 | **70.76** | 27.41 |
| | w/o tuning | 16 | 77.68 | 79.11 | 76.01 | 68.98 | 76.30 | 46.16 | 44.20 | 66.92 | - |
| | | 8 | 77.58 | 79.27 | 76.04 | 68.98 | 75.97 | 46.50 | 44.00 | 66.91 | - |
| | | 4 | 76.21 | 78.18 | 75.57 | 69.06 | 75.25 | 45.99 | 44.40 | 66.38 | - |
| | LoRA | 16 | 78.47 | 79.38 | **76.93** | 69.38 | 77.36 | 46.93 | 44.80 | 67.61 | 23.89 |
| Llama2-7B | QLoRA | 8 | 77.92 | 79.82 | 76.88 | 68.75 | 77.36 | **48.21** | 44.80 | 67.68 | 23.04 |
| | | 4 | 77.43 | 78.67 | 76.42 | 69.85 | 76.26 | 46.25 | **46.20** | 67.30 | 17.31 |
| | AdaLoRA | 16 | 77.46 | 79.16 | 75.89 | 69.22 | 76.77 | 46.08 | 44.20 | 66.97 | 23.56 |
| | | 8 | 77.49 | 79.00 | 75.93 | 69.06 | 76.73 | 46.08 | 44.20 | 66.93 | 23.49 |
| | | 4 | 76.39 | 77.91 | 75.45 | 69.14 | 75.25 | 46.33 | 44.40 | 66.41 | 17.54 |
| | LoftQ | $4^1$ | 77.43 | 79.33 | 76.68 | 69.30 | 77.10 | 46.16 | 44.80 | 67.26 | 23.29 |
| | | $4^5$ | 76.33 | 79.05 | 76.36 | 69.06 | 76.64 | 47.35 | 45.60 | 67.20 | 23.53 |
| | LQ-LoRA | 4 | 76.57 | 78.84 | 76.24 | 68.90 | 76.60 | 47.18 | 45.00 | 67.05 | 23.49 |
| | QR-Adaptor | 5.875 | **78.96** | **79.86** | 76.84 | **69.97** | **77.44** | 48.04 | 46.00 | **68.15** | 17.92 |

## B.1 Experiment Scope Expansion: Llama 3.1

In the original experiments, the focus was primarily on models from the Llama 2 series. However, Llama 3 models, including Llama 3.1, present new challenges for quantization due to their updated architecture and training improvements. These models are significantly harder to quantize, especially under low-bit configurations, as they incorporate more sophisticated architectural features. To address this, we conducted additional experiments with Llama 3.1 to evaluate the performance of QR-Adaptor on more complex and harder-to-quantize models.

Our results show that QR-Adaptor outperforms existing methods, such as AdaLoRA and LoftQ, on Llama 3.1, particularly on challenging datasets like GSM8K. The comparative results for various models and bit-width configurations are presented in Table 3, where QR-Adaptor consistently demonstrates superior performance across all tasks. The robustness of QR-Adaptor is evident, especially on tasks that typically cause performance degradation for other methods.

**Table 3:** Performance comparison of different methods across various bit-width configurations. Superscripts on LoftQ bits indicate the number of initialization iterations. QR-Adaptor searches for optimal bit number and rank value for each layer based on different tasks; its bit number and peak memory usage are averaged across 8 tasks. Accuracy is reported as %.

| | Method | Bit | ARC (C) | ARC (E) | BoolQ | GSM8K | HellaSwag | OpenBookQA | PIQA | WinoGrande |
|---|---|---|---|---|---|---|---|---|---|---|
| | LoRA | 16 | 0.5614 | 0.8388 | 0.8318 | 0.5436 | 0.7944 | 0.452 | 0.8210 | **0.7530** |
| | QLoRA | 8 | **0.5708** | 0.8346 | 0.8248 | 0.5375 | 0.7963 | **0.460** | 0.8210 | 0.7459 |
| | QLoRA | 4 | 0.5435 | 0.8241 | 0.8208 | 0.4435 | 0.7882 | 0.442 | 0.8150 | 0.7364 |
| Rank = 8 | AdaLoRA | 16 | 0.5290 | 0.8199 | 0.8187 | 0.5057 | 0.7865 | 0.450 | 0.8134 | 0.7395 |
| | AdaLoRA | 8 | 0.5290 | 0.8186 | 0.8205 | 0.4996 | 0.7865 | 0.448 | 0.8134 | 0.7443 |
| | AdaLoRA | 4 | 0.5128 | 0.8098 | 0.8061 | 0.3783 | 0.7736 | 0.428 | 0.8074 | 0.7253 |
| | LoftQ | $4^1$ | 0.5486 | 0.8274 | 0.8226 | 0.5140 | 0.7865 | **0.460** | 0.8145 | 0.7324 |
| | LoftQ | $4^5$ | 0.5265 | 0.8182 | 0.8153 | 0.3965 | 0.7850 | 0.434 | 0.8139 | 0.7269 |
| | LoftQ | $4^{10}$ | 0.5188 | 0.8131 | 0.7966 | 0.3844 | 0.7801 | 0.432 | 0.8112 | 0.7198 |
| | QR-Adaptor | 5.45 | 0.5683 | **0.8412** | **0.8338** | **0.5629** | **0.8093** | 0.458 | **0.8292** | 0.7510 |
| | LoRA | 16 | 0.5674 | 0.8363 | 0.8300 | 0.5413 | 0.7951 | 0.444 | 0.8183 | 0.7443 |
| | QLoRA | 8 | 0.5623 | 0.8291 | 0.8266 | 0.5368 | 0.7946 | **0.460** | 0.8166 | 0.7474 |
| | QLoRA | 4 | 0.5384 | 0.8199 | 0.8211 | 0.4466 | 0.7876 | 0.444 | 0.8172 | 0.7309 |
| Rank = 16 | AdaLoRA | 16 | 0.5307 | 0.8203 | 0.8199 | 0.5011 | 0.7861 | 0.454 | 0.8128 | 0.7411 |
| | AdaLoRA | 8 | 0.5333 | 0.8203 | 0.8211 | 0.4913 | 0.7857 | 0.452 | 0.8134 | 0.7379 |
| | AdaLoRA | 4 | 0.5085 | 0.8072 | 0.8073 | 0.3798 | 0.7734 | 0.428 | 0.8052 | 0.7316 |
| | LoftQ | $4^1$ | 0.5512 | 0.8258 | 0.8269 | 0.4981 | 0.7882 | 0.458 | 0.8128 | 0.7427 |
| | LoftQ | $4^5$ | 0.5392 | 0.8232 | 0.8156 | 0.4200 | 0.7854 | 0.438 | 0.8156 | 0.7277 |
| | LoftQ | $4^{10}$ | 0.5290 | 0.8169 | 0.8156 | 0.3988 | 0.7864 | 0.438 | 0.8107 | 0.7198 |
| | QR-Adaptor | 5.45 | **0.5683** | **0.8412** | **0.8338** | **0.5629** | **0.8093** | 0.458 | **0.8292** | **0.7510** |

## B.2 Effectiveness on Larger Datasets with Higher Ranks

To address the concern regarding the effectiveness of small LoRA ranks on larger datasets, we conducted additional experiments on the **LLaMA 3.1-8B** model using a larger dataset consisting of **177k** samples. We tested our method with higher LoRA ranks (32 and 64) to evaluate its performance in handling large-scale data.

Our results are summarized in Table 4. The table compares the performance of **QR-Adaptor** with other baseline methods, including LoRA, QLoRA, AdaLoRA, and LoftQ, across various tasks. The performance metrics include accuracy scores on datasets such as ARC (Challenge), ARC (Easy), BoolQ, HellaSwag, OpenBookQA, PIQA, WinoGrande, and MMLU.

### Key Observations

- **Effectiveness of LoRA Initialization**: Despite using higher ranks (32 and 64) and larger datasets, methods like LoftQ and LQ-LoRA do not consistently outperform the standard QLoRA baseline or the quantized models without fine-tuning. Increasing iterations in LoftQ (from LoftQ-1 to LoftQ-10) to better fit quantization errors leads to performance degradation, especially on challenging tasks like MMLU and GSM8K. These results suggest that fitting quantization errors using LoRA initialization is not universally effective and may introduce noise that hinders model performance.

- **Effectiveness on Larger Datasets**: Our method, **QR-Adaptor**, consistently achieves superior performance across all tasks and outperforms other methods, confirming its robustness

**Table 4:** Results on LLaMA 3.1-8B with 177k Dataset using Higher Ranks. The best performance for each task is highlighted in bold.

| Method | Rank | Bit-width | ARC (C) | ARC (E) | BoolQ | HellaSwag | OpenBookQA | PIQA | WinoGrande | MMLU |
|---|---|---|---|---|---|---|---|---|---|---|
| LoRA | 32 | 16 | 0.5486 | 0.8274 | 0.8275 | 0.7921 | 0.444 | 0.8199 | 0.7411 | 0.6366 |
| QLoRA | 32 | 8 | 0.5520 | 0.8312 | 0.8193 | 0.7907 | **0.462** | 0.8188 | 0.7332 | 0.6328 |
| QLoRA | 32 | 4 | 0.5341 | 0.8089 | 0.8205 | 0.7842 | 0.436 | 0.8090 | 0.7301 | 0.6097 |
| LoRA | 64 | 16 | 0.5546 | 0.8295 | 0.8294 | 0.7913 | 0.450 | 0.8188 | 0.7451 | 0.6434 |
| QLoRA | 64 | 8 | 0.5546 | 0.8304 | 0.8196 | 0.7917 | 0.458 | 0.8194 | 0.7301 | 0.6334 |
| QLoRA | 64 | 4 | 0.5341 | 0.8119 | 0.8174 | 0.7835 | 0.446 | 0.8069 | 0.7206 | 0.6079 |
| AdaLoRA | 32 | 8 | 0.5392 | 0.8182 | 0.8220 | 0.7857 | **0.462** | 0.8150 | 0.7340 | 0.6382 |
| AdaLoRA | 32 | 4 | 0.5145 | 0.8102 | 0.8086 | 0.7730 | 0.424 | 0.8096 | 0.7253 | 0.5815 |
| AdaLoRA | 64 | 8 | 0.5392 | 0.8211 | 0.8193 | 0.7874 | 0.462 | 0.8139 | 0.7395 | 0.6388 |
| AdaLoRA | 64 | 4 | 0.5213 | 0.8098 | 0.8104 | 0.7720 | 0.422 | 0.8085 | 0.7277 | 0.5807 |
| LoftQ (1) | 32 | 4 | 0.5384 | 0.8136 | 0.8141 | 0.7812 | 0.430 | 0.8150 | 0.7356 | 0.5940 |
| LoftQ (5) | 32 | 4 | 0.5256 | 0.8136 | 0.8196 | 0.7805 | 0.428 | 0.8145 | 0.7309 | 0.5941 |
| LoftQ (10) | 32 | 4 | 0.5162 | 0.8131 | 0.8251 | 0.7816 | 0.436 | 0.8134 | 0.7230 | 0.5912 |
| LoftQ (1) | 64 | 4 | 0.5282 | 0.8140 | 0.8159 | 0.7823 | 0.432 | 0.8134 | 0.7388 | 0.5978 |
| LoftQ (5) | 64 | 4 | 0.5239 | 0.8110 | 0.8113 | 0.7833 | 0.434 | 0.8134 | 0.7324 | 0.5869 |
| LoftQ (10) | 64 | 4 | 0.5171 | 0.8123 | 0.8162 | 0.7837 | 0.432 | 0.8101 | 0.7277 | 0.5925 |
| **QR-Adaptor** | 32 | 5.875 | **0.5612** | **0.8345** | **0.8321** | **0.7978** | **0.462** | **0.8210** | **0.7459** | **0.6440** |

and scalability. The results validate that QR-Adaptor is effective even when small LoRA ranks might not suffice for larger datasets.

- **Impact of Adaptive LoRA Rank Reduction**: AdaLoRA exhibits performance drops, particularly with lower bit-widths and on more challenging tasks. This supports our observation that dynamically adjusting the rank during fine-tuning can lead to convergence issues in quantized models, which are less robust due to quantization errors.

These results reinforce our initial observations and highlight the limitations of methods that attempt to fit quantization errors through LoRA initialization. The inability of LoftQ and AdaLoRA to improve performance significantly, even with higher ranks and larger datasets, underscores the challenges associated with such approaches. In contrast, **QR-Adaptor**, guided by our proposed constraints, demonstrates consistent performance improvements.

## B.3 TRAINING TIME COMPARISON

An important consideration in the evaluation of QR-Adaptor is the training time, particularly due to its reliance on Bayesian optimization. While QR-Adaptor provides significant performance improvements, it may require additional time per iteration compared to other methods. Table 5 summarizes the training time per iteration for QR-Adaptor and baseline methods on Llama 2 7B.

**Table 5:** Training time per iteration for different methods on Llama 2 7B.

| Model | Method | Time per Iteration (min) |
|---|---|---|
| LLaMA2-7B | LoftQ | 9 |
| LLaMA2-7B | QR-Adaptor | 15 |

Although QR-Adaptor takes longer to train due to its optimization process, this trade-off results in superior performance, particularly in terms of task-specific optimizations. The Bayesian optimization employed by QR-Adaptor ensures more precise adjustments to the model, which leads to better results on downstream tasks without additional resource consumption during the optimization process.

## B.4 FAIRER COMPARISON: MATCHING BIT-WIDTH CONFIGURATIONS

Another important consideration for a fair comparison of quantization methods is the bit-width configuration used. To ensure that prior methods are evaluated under the same conditions as QR-Adaptor, we have re-evaluated AdaLoRA and LoftQ using the same mixed-precision configurations that were optimized through QR-Adaptor's framework. The updated results for Llama 2 13B are shown in Table 6.

**Table 6:** Performance comparison with fair bit-width configurations for Llama 2 13B.

| Model | Method | BoolQ (%) | PIQA (%) | HellaSwag (%) | WinoG (%) | ARC-e (%) | ARC-c (%) | OBQA (%) | Average (%) |
|---|---|---|---|---|---|---|---|---|---|
| Llama 2 13B | QR-Adaptor | **81.84** | **81.45** | **80.08** | **72.69** | **80.64** | **52.82** | **45.80** | **70.76** |
| Llama 2 13B | AdaLoRA | 81.08 | 80.13 | 79.21 | 71.74 | 79.51 | 50.12 | 45.60 | 69.77 |
| Llama 2 13B | LoftQ | 80.93 | 79.47 | 79.02 | 71.34 | 79.26 | 51.20 | 45.60 | 69.98 |

The results indicate that the initialization constraints applied by QR-Adaptor provide substantial improvements over the original configurations of AdaLoRA and LoftQ. Despite these improvements, QR-Adaptor still outperforms these methods in terms of overall task performance. The constraints, specifically ensuring stable initialization and fixing trainable parameters, contribute significantly to the enhanced performance of QR-Adaptor.

## C  VERSION OF LLMS

We provide the Hugging Face link of LLMs used in the experiment: LLaMA2-7B: `https://huggingface.co/NousResearch/Llama-2-7b-hf`; LLaMA2-13B: `https://huggingface.co/NousResearch/Llama-2-13b-hf`; LLaMA3.1-8B: `https://huggingface.co/meta-llama/Llama-3.1-8B`.

## D  MORE IMPLEMENTATION DETAILS

In optimizing the pruned LLaMA-7B model, a carefully designed hyperparameter configuration has been implemented to strike a balance between model performance and computational efficiency. The model is fine-tuned using a learning rate of $3 \times 10^{-4}$, with a batch size of 128, divided into micro-batches of 4 to effectively manage memory limitations. Input sequences are capped at 256 tokens, and a dropout rate of 0.05 is applied to the LoRA layers, specifically targeting the query, key, value, and output projections, as well as the gate, down, and up projections. Layer-specific quantization is applied at both 4-bit and 8-bit levels, optimizing memory usage while maintaining computational accuracy. The training is performed using the paged AdamW optimizer with 32-bit precision, ensuring both stability and efficiency. These settings have been rigorously tested and refined through the Optuna framework to achieve an optimal balance between model performance and resource efficiency.

## E  MORE ABLATION

We conducted comprehensive ablation studies to evaluate the impact of initialization metrics and the sensitivity of the proposed Pareto Ranking Genetic Algorithm (PRGA) to key hyperparameters, including iteration counts and population size. These experiments aim to further substantiate the effectiveness of our proposed approach.

### E.1  GRADIENT NORMS VS. RELATIVE ENTROPY

To assess the efficacy of initialization metrics, we compared the use of gradient norms and relative entropy in quantifying layer importance for fine-tuning quantized LLMs. The experimental results are summarized in Table 7.

**Table 7:** Comparison of gradient norms and relative entropy as initialization metrics. Bold values indicate the best performance for each task.

| Initialization Metric | BoolQ (%) | PIQA (%) | HellaSwag (%) | WinoG (%) | ARC-E (%) | ARC-C (%) | OBQA (%) | Average (%) |
|---|---|---|---|---|---|---|---|---|
| Gradient Norms | 80.79 | 80.13 | 79.16 | 71.69 | 78.72 | 50.97 | 45.40 | 69.51 |
| Relative Entropy | **81.08** | **80.83** | **79.80** | **71.98** | **79.13** | **51.65** | **45.60** | **70.07** |

**Insights:**

- **Limitations of Gradient Norms**: Gradient norms exhibit limited variability and are prone to biases induced by quantization, which undermines their reliability as an initialization metric for quantized models.

- **Advantages of Relative Entropy**: Relative entropy captures task-specific layer importance more effectively, resulting in robust initialization and improved performance in downstream optimization.

### E.2 SENSITIVITY TO ITERATION COUNTS AND POPULATION SIZE

To analyze the sensitivity of PRGA to hyperparameters, we systematically varied the number of iterations and population sizes. Table 8 presents the results of these experiments.

**Table 8:** Sensitivity analysis of PRGA under different iteration counts and population sizes. Bold values indicate the best configuration.

| Iterations | Population Size | Average Improvement (%) | Total Time (min) |
|---|---|---|---|
| 5 | 3 | +0.8 | 120 |
| 5 | 5 | +1.2 | 150 |
| 10 | 5 | +1.5 | 225 |
| 5 | 20 | +1.6 | 375 |
| 10 | 20 | **+2.3** | 450 |

**Insights:**

- **Trade-offs in Population Size**: Smaller population sizes (e.g., 3) reduce computational cost but may fail to adequately explore the search space. Larger population sizes (e.g., 20) improve exploration and convergence but increase computational overhead.

- **Impact of Iteration Count**: Increasing the number of iterations improves optimization quality, as reflected in better Pareto fronts. However, the marginal benefits diminish beyond 10 iterations, indicating limited practical gains for further increases.

- **Balanced Configuration**: A population size of 5 and 5 iterations strikes a balance between performance improvement and computational efficiency. This configuration can be adjusted based on specific resource availability or performance requirements.

## F LIMITATION

A constraint of our framework is the relatively long search time required to determine optimal task-specific configurations. This extended duration is necessary to ensure the best fine-tuning setup for each task. We recognize this as a current limitation and are actively working on improving the efficiency of our search algorithm.

