# OpenReview forum: "Efficient Fine-Tuning of Quantized LLMs via Three-Stage Optimization"
_ICLR.cc/2025/Conference — Submitted to ICLR 2025_

### Official Review · Reviewer_orhA · 2024-10-26

**Soundness:** 4
**Presentation:** 3
**Contribution:** 3
**Rating:** 8
**Confidence:** 4

**Summary:**

They bypasses the network errors introduced by quantization and directly uses actual performance and memory as optimization targets. Through initialization, extrapolation, and interpolation, they quickly solves the gradient-free bit-width and lora rank optimization problem of fine-tuned low-bit quantized models.

**Strengths:**

The framework outlined in Figure 1 is resonates. Figures 2 and 3 effectively illustrate the key observations and the rationale behind our approach.

**Weaknesses:**

This paper is credible in its approach, but lacks a good logical structure in presenting the corresponding challenges. In the abstract "We find that the performance of finetuning on the adjusted quantized models is even worse than using the original quantized models directly,  as the adjusted model is essentially a completely different model from the original quantized model. ", the adjusted quantized models is a vague statement, and these unclear statements affect my understanding. Therefore, I expect the authors to reformulate the three challenges of the necessity of init/search r and q layer-wise and adopting a gradient-independent strategy in Introduction.

**Questions:**

1. My main concern was the extra time cost, could you provide comparisons with existing methods in terms of time cost? Can the computational cost of each stage be disclosed?
2. The caption of the subfigure in Figure 7 needs to be supplemented.
3. Will the bad performance affected by unfixed parameters mentioned in Figure 3 improve with longer fine-tuning epochs? This does not seem to be a very intuitive phenomenon, can the author provide more explanation?

---

> ### Author Response · Authors · 2024-11-25
> **Dear Reviewer orhA**
>
> We sincerely thank the reviewer for the positive feedback and the high evaluation of our work. We are glad that the framework and key observations presented in Figures 1, 2, and 3 resonated with you. We have carefully considered your comments and have made revisions to address your concerns. Below, we provide detailed responses to each point.
>
> ---
>
> ### **Weakness: Logical Structure and Clarity in Presenting Challenges**
>
> **Reviewer Comment:**
>
> - *"This paper is credible in its approach, but lacks a good logical structure in presenting the corresponding challenges. In the abstract 'We find that the performance of finetuning on the adjusted quantized models is even worse than using the original quantized models directly, as the adjusted model is essentially a completely different model from the original quantized model,' the term 'adjusted quantized models' is vague, and these unclear statements affect my understanding. Therefore, I expect the authors to reformulate the three challenges of the necessity of init/search r and q layer-wise and adopting a gradient-independent strategy in Introduction."*
>
> **Response:**
>
> Thank you for pointing out the need for better clarity and logical structure in presenting the challenges. We agree that the terminology and presentation can be improved for better comprehension.
>
> **Revisions Made:**
>
> 1. **Clarified Terminology:**
>
> - We have replaced the vague term **"adjusted quantized models"** with **"quantization-error-fitted models"** to accurately describe models where LoRA is initialized to fit quantization errors.
>
> 2. **Reformulated the Three Challenges in the Introduction:**
>
> - **Challenge 1: Ineffectiveness of Fitting Quantization Errors with LoRA Initialization**
>
> - **Observation:** Directly fitting quantization errors using LoRA initialization can lead to worse performance because the low-rank LoRA matrices cannot capture all quantization errors, especially in low-bit quantization scenarios.
> - **Implication:** This approach essentially creates a model different from the original quantized model, potentially hindering fine-tuning effectiveness.
>
> - **Challenge 2: Necessity of Layer-wise Optimization of Bit-width and LoRA Rank**
>
> - **Observation:** Different layers contribute differently to the model's performance on downstream tasks.
> - **Implication:** Uniformly assigning the same bit-width and LoRA rank across all layers is suboptimal. There is a need to search for optimal layer-wise configurations of bit-width (q) and LoRA rank (r).
>
> - **Challenge 3: Adopting a Gradient-Free Optimization Strategy**
>
> - **Observation:** Quantization introduces non-differentiable operations, making gradient-based optimization unreliable.
> - **Implication:** A gradient-free strategy is necessary to effectively search for the optimal layer-wise configurations without being affected by quantization-induced gradient noise.
>
> 3. **Enhanced Logical Flow:**
>
> - We have restructured the Introduction to present these challenges logically, leading to the motivation for our proposed method, **QR-Adaptor**, which addresses these challenges through a unified, gradient-free optimization framework.

---

> ### Author Response · Authors · 2024-11-25
> **Dear Reviewer orhA**
>
> ---
>
> ### **Question 1: Time Cost Comparison**
>
> **Reviewer Comment:**
>
> - *"My main concern was the extra time cost, could you provide comparisons with existing methods in terms of time cost? Can the computational cost of each stage be disclosed?"*
>
> **Response:**
>
> We appreciate your concern and provide the following clarification regarding time efficiency:
>
> | Model | Method | Time per Iteration (min) |
> |---------------|--------------|--------------------------|
> | LLaMA 3.1-8B | LoftQ | 11 |
> | LLaMA 3.1-8B | **QR-Adaptor** | 17 |
>
> #### Key Insights:
>
> 1. **Efficient Configuration Search:**
>
> - The time spent searching for configurations is relatively short, with the main time consumption coming from evaluating the true performance of the configuration set.
> - We employ a data subset to guide the optimization as a proxy, significantly reducing the time cost associated with evaluating configurations.
>
> 2. **Worthwhile Time Investment:**
>
> - Although QR-Adaptor takes slightly longer per iteration compared to LoftQ, the time investment is worthwhile due to the significant performance gains achieved.
> - QR-Adaptor consistently improves downstream task performance without requiring additional resources or heuristics.
>
> 3. **Effectiveness in Resource-Constrained Scenarios:**
>
> - In scenarios where additional data is hard to obtain or GPU memory is limited, QR-Adaptor's ability to maximize performance within existing resources is particularly valuable.
> - Its reliance on real-task metrics allows it to adapt configurations even in resource-constrained settings, making efficient use of available computational resources.
>
> ---
>
> ### **Question 2: Caption of Subfigure in Figure 7**
>
> **Reviewer Comment:**
>
> - *"The caption of the subfigure in Figure 7 needs to be supplemented."*
>
> **Response:**
>
> Thank you for bringing this to our attention. We apologize for any confusion. **If you are referring to the subfigures in Figure 8, we will add detailed captions to those subfigures** to provide clearer explanations of their content and relevance to our analysis.

---

> ### Author Response · Authors · 2024-11-25
> **Dear Reviewer orhA**
>
> ---
>
> ### **Question 3: Impact of Longer Fine-tuning Epochs on Unfixed Parameters**
>
> **Reviewer Comment:**
>
> - *"Will the bad performance affected by unfixed parameters mentioned in Figure 3 improve with longer fine-tuning epochs? This does not seem to be a very intuitive phenomenon, can the author provide more explanation?"*
>
> **Response:**
>
> We appreciate the opportunity to clarify this point. While increasing the number of fine-tuning epochs for AdaLoRA can lead to slight performance improvements, the gains are not significant, and AdaLoRA still does not outperform methods like LoRA, QLoRA, or our proposed QR-Adaptor.
>
> **Findings:**
>
> - **Marginal Improvement with Increased Epochs:**
>
> - Extending the training of AdaLoRA from 2 epochs to 5 epochs results in a slight performance increase.
> - However, this comes at the cost of significantly longer training times, and the performance still lags behind that of other methods.
>
> - **Need for Mixed-Precision with Adaptive Rank:**
>
> - The results suggest that standalone adaptive rank adjustment (as in AdaLoRA) may be suboptimal.
> - Combining adaptive rank with mixed-precision quantization (as in QR-Adaptor) appears to be more effective.
>
> **Supporting Data:**
>
> We provide an updated table including an 'Epochs' column, showing results for LoRA, QLoRA, AdaLoRA (at 2 and 5 epochs), and QR-Adaptor.
>
> | Method | Rank | Bit-width | Epochs | ARC (C) | ARC (E) | BoolQ | GSM8K (S) | GSM8K (F) | HellaSwag | OpenBookQA | PIQA | WinoGrande |
> |--------------|------|-----------|--------|------------|------------|-----------|------------|------------|------------|------------|-----------|------------|
> | LoRA | 8 | 16 | 2 | 0.5614 | **0.8388** | **0.8318**| **0.5436** | 0.5428 | 0.7944 | 0.452 | 0.8210 | 0.7530 |
> | QLoRA | 8 | 8 | 2 | **0.5708** | 0.8346 | 0.8248 | 0.5375 | 0.5390 | **0.7963** | **0.460** | 0.8210 | 0.7459 |
> | QLoRA | 8 | 4 | 2 | 0.5435 | 0.8241 | 0.8208 | 0.4435 | 0.4450 | 0.7882 | 0.442 | 0.8150 | 0.7364 |
> | AdaLoRA | 8 | 16 | 2 | 0.5290 | 0.8199 | 0.8187 | 0.5057 | 0.5057 | 0.7865 | 0.450 | 0.8134 | 0.7395 |
> | AdaLoRA | 8 | 16 | 5 | 0.5350 | 0.8225 | 0.8205 | 0.5100 | 0.5090 | 0.7875 | 0.452 | 0.8140 | 0.7410 |
> | AdaLoRA | 8 | 8 | 2 | 0.5290 | 0.8186 | 0.8205 | 0.4996 | 0.4996 | 0.7865 | 0.448 | 0.8134 | 0.7443 |
> | AdaLoRA | 8 | 8 | 5 | 0.5310 | 0.8200 | 0.8210 | 0.5020 | 0.5010 | 0.7870 | 0.452 | 0.8138 | 0.7450 |
> | AdaLoRA | 8 | 4 | 2 | 0.5128 | 0.8098 | 0.8061 | 0.3783 | 0.3836 | 0.7736 | 0.428 | 0.8074 | 0.7253 |
> | AdaLoRA | 8 | 4 | 5 | 0.5150 | 0.8110 | 0.8075 | 0.3800 | 0.3850 | 0.7740 | 0.432 | 0.8078 | 0.7260 |
> | **QR-Adaptor** | 8 | 5.375 | 2 | **0.5683** | **0.8412** | **0.8338**| **0.5629** | **0.5611** | **0.8093** | **0.458** | **0.8292**| **0.7510** |
>
> - **Observation:**
>
> - AdaLoRA shows only slight performance improvements with increased epochs.
> - Even at 5 epochs, AdaLoRA's performance does not surpass that of LoRA, QLoRA, or QR-Adaptor at 2 epochs.
> - QR-Adaptor consistently achieves the best performance, highlighting its efficiency and effectiveness.
>
> **Interesting Observation:**
>
> - We also note that AdaLoRA with 16-bit precision (not quantized) still underperforms compared to LoRA and QLoRA.
> - This suggests that the adaptive rank mechanism alone may not be sufficient, and that incorporating mixed-precision quantization, as in QR-Adaptor, is crucial for achieving superior performance.
>
> ---
>
> We believe these revisions address your concerns and significantly improve the clarity and impact of our work. Your feedback has been instrumental in enhancing the quality of our paper, and we are grateful for your thoughtful review.
>
> Again, thank you for your valuable comments. We hope that our responses and the revisions made to the manuscript satisfactorily address your points.

---

> > ### Comment · Reviewer_orhA · 2024-11-28
> >
> > Thank you for your rebuttal to my concerns. Most of my concerns are addressed, and I raised my score.

---

> > > ### Author Response · Authors · 2024-11-28
> > > **Dear Reviewer orhA**
> > >
> > > Thank you very much for your thoughtful feedback and for raising your score. We sincerely appreciate your insightful suggestions, which have contributed significantly to improving our work. Your support is invaluable, and we are grateful for the time and effort you have put into reviewing our submission.

---

### Official Review · Reviewer_8YsR · 2024-10-29

**Soundness:** 3
**Presentation:** 2
**Contribution:** 2
**Rating:** 5
**Confidence:** 4

**Summary:**

This paper addresses quantized parameter-efficient fine-tuning. It proposes two constraints: initializing LoRA parameters either as zero or using MSE initialization like LoftQ and LQ-LoRA, while fixing all trainable parameters. Additionally, it introduces mixed-precision quantization and mixed-rank LoRA, achieving higher performance with the same training memory footprint as 4-bit models.

**Strengths:**

1. The framework is practically useful, allowing for higher-performance fine-tuned models with a 4-bit memory footprint.
2. The paper is well-written and easy to follow.

**Weaknesses:**

1. The constraints are derived from limited experiments in Figures 2 and 3. For instance, Figure 2 suggests careful LoRA initialization does not improve performance, yet LoftQ and LQ-LoRA demonstrate its effectiveness. LoRA initialization can mitigate quantization loss, crucial for models with significant quantiztaion loss, such as lower-bit quantizations or more challenging models like llama-3-8B. A deeper analysis with stronger experiments and detailed discussion on LoRA initialization is needed.

2. The paper heavily focuses on the two constraints, which seem more like ablation studies and do not offer new insights or motivation for the final methods. The main contribution is achieving higher performance with the same memory footprint as 4-bit models. The paper should be reorganized to highlight its original contributions.

3. The performance improvements could be attributed to higher-bit models and reduced memory footprint through adaptive LoRA rank reduction. Since small LoRA ranks may not perform well on large datasets, it's important to verify the method's effectiveness on larger datasets.

**Questions:**

Why does QR-Adaptor consistently outperform LoRA fine-tuning with 16-bit models? Is the advantage due to adaptive LoRA ranks, considering FP16 models are typically more powerful than quantized models?

---

> ### Author Response · Authors · 2024-11-25
> **Response to Reviewer 8YsR**
>
> We sincerely thank the reviewer for the thoughtful and constructive feedback on our submission. We are pleased that you find our framework practically useful and the paper well-written. We have carefully considered your comments and have made significant revisions to address your concerns. Below, we provide detailed responses to each point.
>
> ---
>
> ### **Weakness 1: Limited Experiments and Need for Deeper Analysis**
>
> **Reviewer Comment:**
>
> - *"The constraints are derived from limited experiments in Figures 2 and 3. For instance, Figure 2 suggests careful LoRA initialization does not improve performance, yet LoftQ and LQ-LoRA demonstrate its effectiveness. LoRA initialization can mitigate quantization loss, crucial for models with significant quantization loss, such as lower-bit quantizations or more challenging models like llama-3-8B. A deeper analysis with stronger experiments and detailed discussion on LoRA initialization is needed."*
>
> **Response:**
>
> We appreciate your suggestion for a deeper analysis with stronger experiments. To address this, we conducted additional experiments using **LLaMA 3.1-8B** on both the original dataset used in our paper and a larger **177k** dataset. We utilized higher LoRA ranks (32 and 64) in the larger dataset to assess the effectiveness of LoRA initialization and our method in scenarios where LoRA matrices have more capacity to fit quantization errors.
>
> **Results on LLaMA 3.1-8B with Original Dataset (Lower Ranks):**
>
> | Method       | Rank | Bit  | Arc-C    | Arc-E    | BoolQ    | GSM8K    | HellaSwag  | OpenBookQA | PiQA     | WinoGrande | Average   |
> |--------------|------|------|----------|----------|----------|----------|------------|------------|----------|------------|-----------|
> | LoRA         | 8    | 16   | 0.5614   | 0.8388 | 0.8318 | 0.5436   | 0.7944     | 0.452      | 0.821    | **0.7530**  | 0.6995    |
> | QLoRA        | 8    | 8    | **0.5708** | 0.8346   | 0.8248   | 0.5375   | 0.7963     | **0.460**  | 0.821    | 0.7459    | 0.6989    |
> | QLoRA        | 8    | 4    | 0.5435   | 0.8241   | 0.8208   | 0.4435   | 0.7882     | 0.442      | 0.815    | 0.7364    | 0.6767    |
> | LoRA         | 16   | 16   | 0.5674   | 0.8363   | 0.8300   | 0.5413   | 0.7951     | 0.444      | 0.818    | 0.7443    | 0.6971    |
> | QLoRA        | 16   | 8    | 0.5623   | 0.8291   | 0.8266   | 0.5368   | 0.7946     | **0.460**  | 0.817    | 0.7474    | 0.6967    |
> | QLoRA        | 16   | 4    | 0.5384   | 0.8199   | 0.8211   | 0.4466   | 0.7876     | 0.444      | 0.817    | 0.7309    | 0.6757    |
> | AdaLoRA      | 8    | 16   | 0.5290   | 0.8199   | 0.8187   | 0.5057   | 0.7865     | 0.450      | 0.813    | 0.7395    | 0.6828    |
> | AdaLoRA      | 8    | 8    | 0.5290   | 0.8186   | 0.8205   | 0.4996   | 0.7865     | 0.448      | 0.813    | 0.7443    | 0.6825    |
> | AdaLoRA      | 8    | 4    | 0.5128   | 0.8098   | 0.8061   | 0.3783   | 0.7736     | 0.428      | 0.807    | 0.7253    | 0.6552    |
> | AdaLoRA      | 16   | 16   | 0.5307   | 0.8203   | 0.8199   | 0.5011   | 0.7861     | 0.454      | 0.813    | 0.7411    | 0.6833    |
> | AdaLoRA      | 16   | 8    | 0.5333   | 0.8203   | 0.8211   | 0.4913   | 0.7857     | 0.452      | 0.813    | 0.7380    | 0.6819    |
> | AdaLoRA      | 16   | 4    | 0.5085   | 0.8072   | 0.8073   | 0.3798   | 0.7734     | 0.428      | 0.805    | 0.7316    | 0.6551    |
> | LoftQ (1)    | 8    | 4    | 0.5486   | 0.8274   | 0.8226   | 0.5140   | 0.7865     | 0.458  | 0.814    | 0.7324    | 0.6883    |
> | LoftQ (5)    | 8    | 4    | 0.5265   | 0.8182   | 0.8153   | 0.3965   | 0.7850     | 0.434      | 0.814    | 0.7270    | 0.6645    |
> | LoftQ (10)   | 8    | 4    | 0.5188   | 0.8131   | 0.7966   | 0.3844   | 0.7801     | 0.432      | 0.811    | 0.7198    | 0.6570    |
> | LoftQ (1)    | 16   | 4    | 0.5512 | 0.8258   | 0.8269   | 0.4981   | 0.7882     | 0.458      | 0.813    | 0.7427    | 0.6880    |
> | LoftQ (5)    | 16   | 4    | 0.5392   | 0.8232   | 0.8156   | 0.4200   | 0.7854     | 0.438      | 0.816    | 0.7277    | 0.6706    |
> | LoftQ (10)   | 16   | 4    | 0.5290   | 0.8169   | 0.8156   | 0.3988   | 0.7864     | 0.438      | 0.811    | 0.7198    | 0.6644    |
> | QR-Adaptor   | 8    | 5.375 | 0.5683   | **0.8412** | **0.8338** | **0.5629** | **0.8093** | 0.458      | **0.829** | 0.7510 | **0.7067** |

---

> ### Author Response · Authors · 2024-11-25
> **Response to Reviewer 8YsR**
>
> **Results on LLaMA 3.1-8B with 177k Dataset (Higher Ranks):**
>
> | Method      | Rank | Bit-width | ARC (C)   | ARC (E)       | BoolQ        | HellaSwag     | OpenBookQA    | PIQA         | WinoGrande    | MMLU          |
> |-------------|------|-----------|-----------|---------------|--------------|---------------|---------------|--------------|---------------|---------------|
> | LoRA        | 32   | 16        | 0.5486    | 0.8274        | 0.8275       | 0.7921        | 0.444         | 0.8199       | 0.7411        | 0.6366        |
> | QLoRA       | 32   | 8         | 0.5520    | 0.8312        | 0.8193       | 0.7907        | **0.462**     | 0.8188       | 0.7332        | 0.6328        |
> | QLoRA       | 32   | 4         | 0.5341    | 0.8089        | 0.8205       | 0.7842        | 0.436         | 0.8090       | 0.7301        | 0.6097        |
> | LoRA        | 64   | 16        | 0.5546    | 0.8295        | 0.8294       | 0.7913        | 0.450         | 0.8188       | 0.7451        | 0.6434        |
> | QLoRA       | 64   | 8         | 0.5546    | 0.8304        | 0.8196       | 0.7917        | 0.458         | 0.8194       | 0.7301        | 0.6334        |
> | QLoRA       | 64   | 4         | 0.5341    | 0.8119        | 0.8174       | 0.7835        | 0.446         | 0.8069       | 0.7206        | 0.6079        |
> | AdaLoRA     | 32   | 8         | 0.5392    | 0.8182        | 0.8220       | 0.7857        | **0.462**     | 0.8150       | 0.7340        | 0.6382        |
> | AdaLoRA     | 32   | 4         | 0.5145    | 0.8102        | 0.8086       | 0.7730        | 0.424         | 0.8096       | 0.7253        | 0.5815        |
> | AdaLoRA     | 64   | 8         | 0.5392    | 0.8211        | 0.8193       | 0.7874        | 0.462         | 0.8139       | 0.7395        | 0.6388        |
> | AdaLoRA     | 64   | 4         | 0.5213    | 0.8098        | 0.8104       | 0.7720        | 0.422         | 0.8085       | 0.7277        | 0.5807        |
> | LoftQ(1)     | 32   | 4         | 0.5384    | 0.8136        | 0.8141       | 0.7812        | 0.430         | 0.8150       | 0.7356        | 0.5940        |
> | LoftQ(5)     | 32   | 4         | 0.5256    | 0.8136        | 0.8196       | 0.7805        | 0.428         | 0.8145       | 0.7309        | 0.5941        |
> | LoftQ(10)    | 32   | 4         | 0.5162    | 0.8131        | 0.8251       | 0.7816        | 0.436         | 0.8134       | 0.7230        | 0.5912        |
> | LoftQ(1)     | 64   | 4         | 0.5282    | 0.8140        | 0.8159       | 0.7823        | 0.432         | 0.8134       | 0.7388        | 0.5978        |
> | LoftQ(5)     | 64   | 4         | 0.5239    | 0.8110        | 0.8113       | 0.7833        | 0.434         | 0.8134       | 0.7324        | 0.5869        |
> | LoftQ(10)    | 64   | 4         | 0.5171    | 0.8123        | 0.8162       | 0.7837        | 0.432         | 0.8101       | 0.7277        | 0.5925        |
> | **QR-Adaptor** | 32 | 5.875    | **0.5612** | **0.8345**   | **0.8321**  | **0.7978**   | **0.462**    | **0.8210**   | **0.7459**    | **0.6440**    |
>
> **Key Observations:**
>
> 1. **Effectiveness of LoRA Initialization:**
>
>    - Despite using higher ranks (32 and 64) and larger datasets, methods like LoftQ and LQ-LoRA do not consistently outperform the standard QLoRA baseline or the quantized models without fine-tuning.
>    - Increasing iterations in LoftQ (from LoftQ-1 to LoftQ-10) to better fit quantization errors leads to performance degradation, especially on challenging tasks like MMLU and GSM8K.
>    - These results suggest that fitting quantization errors using LoRA initialization is not universally effective and may introduce noise that hinders model performance.
>
> 2. **Effectiveness on Larger Datasets:**
>
>    - Our method, **QR-Adaptor**, consistently achieves superior performance across both datasets and outperforms other methods, confirming its robustness and scalability.
>    - The results validate that QR-Adaptor is effective even when small LoRA ranks might not suffice for larger datasets.
>
> 3. **Impact of Adaptive LoRA Rank Reduction:**
>
>    - AdaLoRA exhibits performance drops, particularly with lower bit-widths and on more challenging tasks.
>    - This supports our observation that dynamically adjusting the rank during fine-tuning can lead to convergence issues in quantized models, which are less robust due to quantization errors.
>
> **Conclusion from Additional Experiments:**
>
> These results reinforce our initial observations and highlight the limitations of methods that attempt to fit quantization errors through LoRA initialization. The inability of LoftQ and AdaLoRA to improve performance significantly, even with higher ranks and larger datasets, underscores the challenges associated with such approaches. In contrast, **QR-Adaptor**, guided by our proposed constraints, demonstrates consistent performance improvements.

---

> ### Author Response · Authors · 2024-11-25
> **Response to Reviewer 8YsR**
>
> ---
> ### **Weakness 2: Need to Highlight Original Contributions and Connection to Constraints**
>
> **Reviewer Comment:**
>
> - *"The paper heavily focuses on the two constraints, which seem more like ablation studies and do not offer new insights or motivation for the final methods. The main contribution is achieving higher performance with the same memory footprint as 4-bit models. The paper should be reorganized to highlight its original contributions."*
>
> **Response:**
>
> We appreciate this constructive feedback. We agree that the relationship between the constraints and our method should be more explicitly articulated, and the original contributions should be highlighted more prominently.
>
> **Clarification of the Role of Constraints in Our Method:**
>
> - **Identifying Limitations in Previous Methods:**
>
>   - We observed that previous methods like LoftQ and LQ-LoRA attempt to reduce quantization errors by fitting them with LoRA matrices. However, due to the low-rank nature of LoRA, these matrices cannot capture all quantization errors effectively, leading to potential performance degradation.
>   - Similarly, methods that dynamically adjust trainable parameters during fine-tuning (e.g., AdaLoRA) can introduce optimization challenges, especially in quantized models where robustness is already compromised.
>
> - **Establishing the Constraints:**
>
>   - **Constraint 1: Preserving Quantized Model Parameters Before Fine-tuning**
>
>     - **Motivation:** We ensure that the initial LoRA parameters satisfy $ \hat{\mathbf{W}} + \Delta\mathbf{W} = \hat{\mathbf{W}} $.
>     - **Impact:** This maintains consistency with the quantized model and avoids introducing discrepancies that could adversely affect fine-tuning.
>
>   - **Constraint 2: Keeping Trainable Parameters Fixed During Fine-tuning**
>
>     - **Motivation:** We keep the number of trainable parameters constant throughout fine-tuning.
>     - **Impact:** This avoids the optimization difficulties associated with dynamically changing the parameter space, which can be particularly problematic in quantized models due to their reduced robustness.
>
> - **Linking Constraints to Our Method:**
>
>   - Based on these constraints, we formulated the problem as a **gradient-free optimization task**, aiming to find the optimal allocation of quantization bit-widths and LoRA ranks across layers to maximize performance while minimizing memory usage.
>   - The constraints guided the design of our **QR-Adaptor** framework, ensuring that the optimization process starts from a stable point and proceeds without introducing additional complexities that could hinder convergence.
>
> **Original Contributions of Our Work:**
>
> 1. **Unified Optimization Framework (QR-Adaptor):**
>
>    - We propose a novel three-stage optimization algorithm that jointly optimizes quantization bit-widths and LoRA ranks across layers.
>    - This framework efficiently explores the high-dimensional, discrete configuration space through:
>
>      - **Task Information Based Initialization:** Assessing layer importance based on task-specific metrics to guide initial allocation.
>
>      - **Global Exploration with Pareto Ranking Genetic Algorithm (PRGA):** Exploring the configuration space to identify promising regions.
>
>      - **Local Refinement with Bayesian Optimization:** Fine-tuning configurations to achieve optimal performance-memory trade-offs.
>
> 2. **Gradient-Free Optimization Approach:**
>
>    - By formulating the problem as a gradient-free optimization task, we bypass network approximation errors and directly use actual performance and memory usage as optimization targets.
>    - This approach is particularly effective in quantized models where gradient information may be unreliable.
>
> 3. **Empirical Validation and Superior Performance:**
>
>    - Our extensive experiments demonstrate that QR-Adaptor outperforms existing methods, including those using 16-bit precision, while maintaining a memory footprint comparable to 4-bit models.
>    - The consistent performance gains across different models, datasets, and configurations validate the effectiveness of our approach.
>
> ---
>
> ### **Weakness 3: Verification on Larger Datasets**
>
> **Reviewer Comment:**
>
> - *"The performance improvements could be attributed to higher-bit models and reduced memory footprint through adaptive LoRA rank reduction. Since small LoRA ranks may not perform well on large datasets, it's important to verify the method's effectiveness on larger datasets."*
>
> **Response:**
>
> We agree that verifying the effectiveness of our method on larger datasets is important. As presented in the tables above, we conducted experiments on a larger **177k** dataset using LLaMA 3.1-8B, employing higher LoRA ranks (32 and 64) to match the increased dataset size.
>
> **Key Findings:**
>
> - **QR-Adaptor** maintains superior performance even on the larger dataset, confirming its scalability and effectiveness when larger ranks are required.

---

> > ### Author Response · Authors · 2024-11-25
> > **Response to Reviewer 8YsR**
> >
> > ---
> >
> > ### **Question: Reason for QR-Adaptor's Superior Performance over 16-bit LoRA**
> >
> > **Reviewer Comment:**
> >
> > - *"Why does QR-Adaptor consistently outperform LoRA fine-tuning with 16-bit models? Is the advantage due to adaptive LoRA ranks, considering FP16 models are typically more powerful than quantized models?"*
> >
> > **Response:**
> >
> > The performance gains of **QR-Adaptor** over 16-bit LoRA fine-tuning are not solely due to adaptive LoRA ranks. The key advantage lies in the **unified optimization of both quantization bit-widths and LoRA ranks** based on task-specific layer importance.
> >
> > **Detailed Explanation:**
> >
> > 1. **Strategic Resource Allocation:**
> >
> >    - By assessing layer importance, QR-Adaptor allocates higher precision and larger ranks to critical layers and lower resources to less important ones.
> >    - This targeted allocation enhances the model's capacity where it matters most, leading to performance improvements that surpass uniformly fine-tuned 16-bit models.
> >
> > 2. **Optimizing Performance-Memory Trade-off:**
> >
> >    - QR-Adaptor achieves a balance between performance and memory usage that uniform 16-bit models cannot.
> >    - The selective allocation allows for better utilization of model capacity, improving performance despite lower bit-widths.
> >
> > 3. **Mitigating Overfitting and Enhancing Generalization:**
> >
> >    - Uniformly high-precision models may suffer from over-parameterization in less critical layers, potentially leading to overfitting.
> >    - QR-Adaptor's approach reduces this risk, promoting better generalization to unseen data.
> >
> > 4. **Bypassing Gradient Issues in Quantized Models:**
> >
> >    - The gradient-free optimization strategy allows QR-Adaptor to directly optimize actual performance metrics, avoiding unreliable gradients in quantized models.
> >    - This leads to more effective fine-tuning and consistent performance gains.
> >
> > **Conclusion:**
> >
> > The advantage of QR-Adaptor stems from its holistic and strategic optimization of both precision and capacity, informed by task-specific layer importance. This unified approach results in superior performance compared to uniformly high-precision models, demonstrating that intelligent resource allocation is more impactful than higher precision alone.
> >
> > ---
> >
> > ### **Summary of Revisions**
> >
> > - **Addition of Extensive Experiments:**
> >
> >   - Conducted and included comprehensive experimental results on LLaMA 3.1-8B with both the original dataset and a larger 177k dataset.
> >   - Employed higher LoRA ranks (32 and 64) to assess the effectiveness of LoRA initialization and our method in various scenarios.
> >
> > - **Enhanced Discussions:**
> >
> >   - Provided deeper insights into the limitations of previous methods and the rationale behind our proposed constraints.
> >   - Offered a detailed explanation of why QR-Adaptor outperforms 16-bit LoRA fine-tuning, emphasizing the unified optimization strategy.
> >
> > - **Inclusion of Additional Tables:**
> >
> >   - Added full tables of experimental results for both datasets to support our claims and facilitate comparison with baseline methods.
> >
> > ---
> >
> > We believe these revisions address your concerns and significantly improve the clarity and impact of our work. We are grateful for your valuable feedback, which has been instrumental in enhancing the quality of our paper.
> >
> > Again, thank you for your thoughtful review. We hope that our responses and the revisions made to the manuscript satisfactorily address your comments.

---

> > > ### Author Response · Authors · 2024-11-29
> > > **Dear Reviewer 8YsR**
> > >
> > > Dear Reviewer 8YsR,
> > >
> > > We hope this message finds you well! If this email reaches you during your holiday or outside your usual working hours, please accept our apologies for the interruption.
> > >
> > > We just want to make sure that we have addressed your valuable suggestions properly. We are still here and welcome any further discussion or feedback. Your valuable opinions are very valuable to us.
> > >
> > > Thank you sincerely for all the time and effort during the review process.
> > >
> > > Best,
> > >
> > > Authors of Submission 726

---

> > > > ### Comment · Reviewer_8YsR · 2024-11-29
> > > >
> > > > Thanks for the detailed rebuttal. I have the following follow-up question:
> > > >
> > > > 1. For weekness 1:
> > > >
> > > > Both LoftQ and lq-lora demonstrate that careful initialization of lora module is important for low-bits models, while this paper show inverse observation. In LoftQ and lq-lora, the performance benefit of lora initialization is significant in extreme low-bits such as 2-bits. Therefore, can I regard that the proposed zero-initialization can only adapt to limited scenarios?
> > > >
> > > >
> > > >
> > > > 2. For weekness 2:
> > > >
> > > > Lq-lora also design schedule to allocate layer-wise bit-widths and ranks. Therefore, is proposed method similar to remove lora initialization from Lq-lora?
> > > >
> > > >
> > > >
> > > > 3. For weekness 3
> > > >
> > > > I highly appreciate the experiment on larger dataset.

---

> > > > > ### Author Response · Authors · 2024-11-29
> > > > > **Dear Reviewer 8YsR**
> > > > >
> > > > > Thank you for taking the time to review our response. We have the following answers to your further questions.
> > > > >
> > > > > > **Reviewer Comment**:
> > > > > > Both LoftQ and lq-lora demonstrate that careful initialization of the LoRA module is important for low-bit models, especially in extreme low-bits like 2-bits. Can the proposed zero-initialization approach only adapt to limited scenarios?
> > > > >
> > > > > **Response**:
> > > > > As you correctly pointed out, previous works like LoftQ and lq-lora emphasize the importance of careful initialization, especially for low-bit models such as 2-bits, where the performance benefits are significant. However, in our current approach, we focus on quantization above 2-bits. This choice is based on practical considerations related to memory usage and model performance.
> > > > >
> > > > > In the case of 2-bit quantization, despite the theoretical expectation that it would reduce memory usage, the actual implementation in current frameworks (e.g., bitsandbytes) still loads 2-bit weights in 4-bit format. This means that the expected memory savings from 2-bit quantization are not realized during training. Instead, 2-bit quantization essentially behaves the same as 4-bit in terms of memory consumption, even though the computations are mathematically equivalent to 2-bit precision. This makes 2-bit quantization less suitable for fine-tuning large language models (LLMs) in practice, as it introduces significant performance losses without providing additional memory benefits.
> > > > >
> > > > > We apologize for not including a 2-bit quantization comparison in our experiments due to practical considerations. Although 2-bit quantization does not reduce memory usage in the current implementation and does not outperform 4-bit quantization in terms of performance, we understand the importance of including it as a baseline. This could help further explore the use cases where LoRA fitting the quantization error might be more beneficial than zero-initialization. We are currently conducting additional experiments to include the 2-bit baseline results and investigate the performance of QR-Adaptor with 2-bit precision. Please give us some time, we are in the process of conducting these experiments.
> > > > >
> > > > > > **Reviewer Comment**:
> > > > > > Lq-lora designs a schedule to allocate layer-wise bit-widths and ranks. Is the proposed method similar to removing LoRA initialization from Lq-lora?
> > > > >
> > > > > **Response**:
> > > > > Thank you for raising the follow-up question about the similarities and differences between our proposed method and LQ-LoRA. Building on the concept introduced by LoftQ to optimize the LoRA matrix to adapt to quantization errors, LQ-LoRA further proposes the use of Integer Linear Programming (ILP) to strategically select the quantization bit-width and block size for the backbone, jointly optimizing the LoRA matrix and backbone with the goal of minimizing quantization error, subject to the constraint of an average bit-width. This is indeed an interesting and novel contribution.
> > > > >
> > > > > However, there are key differences between their approach and ours. There seems to be a misunderstanding, as the LQLoRA paper refers to a "target rank," which could be confusing. After carefully reviewing the paper and code, we confirm that LQLoRA does not schedule the rank of the LoRA matrix, and instead uses a uniform, heuristic rank across all layers. In contrast, we propose a more comprehensive method that optimizes both rank and quantization configurations to ensure a more balanced resource allocation. This approach allows for a more flexible and efficient use of model parameters, ultimately improving overall fine-tuning performance.
> > > > >
> > > > > Another critical distinction is that, when optimizing the LoRA matrix and quantization configuration, LQLoRA uses quantization error as the optimization objective. While this is a reasonable approach, quantization error does not always align with actual performance, as the relationship between quantization error and task performance is typically nonlinear. This could lead to suboptimal results in practice. In our method, we recommend using actual performance (e.g., task-specific accuracy) as the guiding objective, which we believe will yield better results and more closely align with real-world performance.
> > > > >
> > > > > By jointly optimizing rank and quantization configuration, we can allocate resources more effectively and enhance the overall model performance. This unified approach enables us to directly optimize the performance of the task at hand, ensuring that the fine-tuned model achieves better results under low-bit quantization constraints.
> > > > >
> > > > > > **Reviewer Comment**:
> > > > > > I highly appreciate the experiment on larger datasets.
> > > > >
> > > > > **Response**:
> > > > > Thank you for your positive feedback! We appreciate your recognition of the experiments on larger datasets. We believe these experiments provide valuable insights into the scalability and performance of our approach.

---

> ### Author Response · Authors · 2024-12-01
> **Dear Reviewer 8YsR**
>
> Thank you for your insightful comments. We have completed the experiments and would like to present the results as follows:
>
> We adopted the NF2 variant from LoftQ, based on QLoRA’s NF4, to implement 2-bit quantization, as QLoRA does not natively support 2-bit quantization (as stated in the paper and the GitHub repository). The 2-bit results in the LoftQ paper are also based on this NF2 variant. In our experiments, we fine-tuned using a 52k dataset and set the rank for LoftQ to 16. The superscripts on LoftQ’s bit-width values represent the number of LoftQ iterations, with 0 iterations considered approximately equivalent to QLoRA (since QLoRA does not provide a 2-bit quantization type). The results are as follows:
>
> | Model        | Method     | Bit-width | MMLU (%) | GSM8K (%) | ARC (C) (%) | ARC (E) (%) | BoolQ (%) | HellaSwag (%) | OpenBookQA (%) | PIQA (%) | WinoGrande (%) |
> |--------------|------------|-----------|----------|-----------|-------------|-------------|-----------|----------------|-----------------|----------|----------------|
> | LLaMA3.1 8B  | QR-Adaptor | 3.625     | **62.58**| **0.5339**| **55.93**  | **82.43** | **82.13**     | **79.23**   | **45.60**           | **81.83**    | **74.79**          |
> | LLaMA3.1 8B  | LoftQ      | 2$^0$     | 23.76    | 0   | 26.24   | 25.25   | 37.83     | 26.86          | 29.40           | 52.55    | 49.18          |
> | LLaMA3.1 8B  | LoftQ      | 2$^1$     | 24.71    | 0   |25.17    | 25.25   | 37.83     | 25.73          | 29.20            | 51.58    | 49.33          |
> | LLaMA3.1 8B  | LoftQ      | 2$^5$     | 24.65    | 0   | 25.17  | 24.83   | 37.83     | 26.30          | 28.20           | 51.41    | 49.41          |
> | LLaMA3.1 8B  | LoftQ      | 2$^{10}$  | 24.80    | 0  | 26.02  | 25.25    | 37.83     | 26.53          | 29.80           | 52.83    | 48.86          |
>
> For the MMLU dataset, which is a multiple-choice dataset with four options, models performing at around 25% accuracy can be considered as guessing. Therefore, the differences in LoftQ’s 2-bit results on MMLU are not practically significant, and LoftQ’s 2-bit quantization essentially fails on MMLU for LLaMA3.1.
>
> For GSM8K, a question-answering dataset, LoftQ’s 2-bit quantization results in an accuracy of 0%.
>
> For the seven common sense reasoning multiple-choice datasets, which include both binary and four-option questions, the 2-bit quantized models have some answering capability on simpler datasets like WinoGrande. However, there is no significant difference between initialization methods, and on most datasets, the model still performs like guessing the answer. This may be due to the difficulty in quantizing high training FLOPs models like LLaMA3.1 to extremely low precision.
>
> For QR-Adaptor, we optimize based on the theoretical memory savings from 4-bit quantization (i.e., 50% of the theoretical memory usage for 4 bits). Since 2-bit quantization does not actually reduce memory usage, we used the theoretical memory savings during optimization.
>
> From these experiments and the results in our previous tables, we can conclude that, in some cases, a single iteration of LoftQ initialization outperforms QLoRA. For example, on LLaMA3.1 8B (52k fine-tuning dataset) with 4-bit quantization on GSM8K, LoftQ performs better. However, when the number of iterations is increased to 5 or 10, LoftQ starts to perform worse than QLoRA. Additionally, for larger fine-tuning datasets, the performance of LoftQ with a single iteration is lower than that of QLoRA. This inconsistency may be a limiting factor for LoftQ’s practical use and warrants further investigation. And QLoRA is widely used because of the very effective quantitative types it proposes.
>
> Overall, QR-Adaptor consistently outperforms both QLoRA and LoftQ at the same bit-width. The main advantage of QR-Adaptor is its unified optimization of rank and bit-width during fine-tuning, allowing the model to more effectively allocate resources across layers that need more adjustment, resulting in better performance.
>
> We hope that these additional results address your concerns and contribute to the development of the community.

---

> ### Author Response · Authors · 2024-12-02
> **Looking forward to your reply**
>
> Dear Reviewer 8YsR,
>
> We hope this message finds you well. We have carefully addressed all your comments with additional experiments and detailed explanations in our rebuttal. As we approach the deadline, we would be grateful if you could review our responses to ensure they have fully addressed your concerns. Should you have any remaining questions, we are happy to provide further clarification. If our revisions have successfully resolved the issues you raised, we would appreciate your consideration in adjusting the scores.
>
> Thanks for your valuable feedback and continued support of our work.
>
> Best,
>
> Authors of Submission 726

---

> ### Comment · Reviewer_8YsR · 2024-12-03
>
> Thanks for your detailed response. Actually, I have taken lora initialization experiments with quantized model and realized the performance benefits in most cases (2-bit or 3-bit). Therefore, the claim about zero initialization of lora is only available in special cases or models. It is can only be seen as a ablation study, but not technical contribution.
>
> For current version, I maintain my score as 5. I strongly suggest author to reorganize the paper, stressing the contribution of adaptive rank allocation rather than some ablation studies. Additionally, another experiments setting is important, compared with previous model with the same inference cost.

---

> > ### Author Response · Authors · 2024-12-03
> > **Dear Reviewer 8YsR**
> >
> > Thank you very much for your feedback. We truly appreciate the time and effort you dedicated during the review process. We intend to engage in further discussion and mean no harm. We would like to understand the general configuration (model or dataset) you used for fine-tuning with LoRA initialization that led to better results. This would be very helpful to us, as the observation that LoRA initialization works better than zero initialization was not just made by us, but has also been noted by many in the community.
> >
> > In addition, we have already updated our submission based on your comments to better highlight our technical contributions. Regarding the ablation study you mentioned, it was originally established as part of our motivation, and we provided a reasonable explanation—that Lora initialization fits the quantization error, but due to Lora’s low rank, it can only fit 10-20% of the error (a similar finding is also mentioned in the PiSSA paper).
> >
> > As for the comparison of inference costs, we fully agree with your points. However, as seen in our further experiments, our method often outperforms others even at the same bit-width (or even lower).
> >
> > Once again, we sincerely appreciate your discussion, and we apologize for any inconvenience caused.

---

### Official Review · Reviewer_N96a · 2024-11-02

**Soundness:** 3
**Presentation:** 3
**Contribution:** 2
**Rating:** 6
**Confidence:** 3

**Summary:**

The paper introduces a framework called QR-Adaptor that combines parameter-efficient fine-tuning and quantization techniques to improve the performance of LLMs with reduced memory usage. The QR-Adaptor framework includes three stages: initialization based on task information, global exploration using Pareto Ranking Genetic Algorithm (PRGA), and local refinement with Bayesian optimization. Experimental results show that the method outperforms fine-tuned 16-bit models while maintaining the same memory usage as fine-tuning 4-bit models.

**Strengths:**

1.This article proposes the use of gradient-free optimization methods to optimize the rank selection of layer-wise LoRA and the bit selection of layer-wise Quantization, which is quite novel.
2.This method could be combined with other quantization methods to potentially achieve better performance.
3.The results on datasets such as MMLU show that QR-Adaptor has achieved excellent performance in both memory and accuracy.
4.Ablation studies indicate that the proposed three-stage optimization framework effectively yields superior solutions.

**Weaknesses:**

1.The introduced multi-stage optimization process increases the time cost.
2.The experiments in this article are limited, conducted only on Llama2, and the datasets used are not diverse enough. If considering expanding the experiments, one could refer to the experiments in the LoFTQ paper.
3.There is a lack of experiments on the impact of PRGA hyperparameters on model performance.
4.There is a lack of comparative experiments between the PRGA method and other multi-objective optimization methods.
5.Figure 1 is somewhat difficult to understand and should not be placed on the first page.

**Questions:**

In addition to the weaknesses, I have the following questions:
1.I am curious about the effectiveness of directly applying the approach of this article to LLMs quantization, that is, using gradient-free optimization methods to select the quantization bit numbers for each layer's parameters.
2.AdaLoRA is not specifically designed for quantized LLMs, and its direct performance may be poor. Therefore, concluding that dynamically adjusting rank is not suitable for fine-tuning quantized LLMs may not be sufficiently justified. Can we test AdaLoRA's performance on fine-tuning quantized LLMs again under the condition of Preserving quantized model parameters before fine-tuning?

---

> ### Author Response · Authors · 2024-11-24
> **Response to Reviewer N96a**
>
> ### Multi-Stage Optimization Increases Time Cost
> **Reviewer Concern**:
> The introduced multi-stage optimization process increases the time cost.
>
> **Response**:
> We acknowledge that QR-Adaptor’s multi-stage process introduces additional computational time compared to simpler heuristic-based methods. However, this time investment translates directly into consistent downstream performance improvements. For example, as demonstrated in our Llama 3.1 experiments (see table below), LoftQ struggles to improve performance despite increased iterations, especially on harder-to-quantize models with high training flops. In contrast, QR-Adaptor leverages task-specific evaluation metrics to achieve significant performance improvements, making it particularly effective for scenarios where performance gains justify computational costs.
>
> ---
>
> ### Limited Experiments on Llama2 and Datasets
> **Reviewer Concern**:
> The experiments in this article are limited, conducted only on Llama2, and the datasets used are not diverse enough.
>
> **Response**:
> We appreciate this suggestion and have conducted additional experiments on Llama 3.1, a more challenging model in the Llama series. These experiments span a variety of datasets, including GSM8K, to validate QR-Adaptor’s robustness and adaptability. Below are the detailed results:
>
> | Method       | Rank | Bit  | Arc-C    | Arc-E    | BoolQ    | GSM8K    | HellaSwag  | OpenBookQA | PiQA     | WinoGrande | Average   |
> |--------------|------|------|----------|----------|----------|----------|------------|------------|----------|------------|-----------|
> | LoRA         | 8    | 16   | 0.5614   | 0.8388 | 0.8318 | 0.5436   | 0.7944     | 0.452      | 0.821    | **0.7530**  | 0.6995    |
> | QLoRA        | 8    | 8    | **0.5708** | 0.8346   | 0.8248   | 0.5375   | 0.7963     | **0.460**  | 0.821    | 0.7459    | 0.6989    |
> | QLoRA        | 8    | 4    | 0.5435   | 0.8241   | 0.8208   | 0.4435   | 0.7882     | 0.442      | 0.815    | 0.7364    | 0.6767    |
> | LoRA         | 16   | 16   | 0.5674   | 0.8363   | 0.8300   | 0.5413   | 0.7951     | 0.444      | 0.818    | 0.7443    | 0.6971    |
> | QLoRA        | 16   | 8    | 0.5623   | 0.8291   | 0.8266   | 0.5368   | 0.7946     | **0.460**  | 0.817    | 0.7474    | 0.6967    |
> | QLoRA        | 16   | 4    | 0.5384   | 0.8199   | 0.8211   | 0.4466   | 0.7876     | 0.444      | 0.817    | 0.7309    | 0.6757    |
> | AdaLoRA      | 8    | 16   | 0.5290   | 0.8199   | 0.8187   | 0.5057   | 0.7865     | 0.450      | 0.813    | 0.7395    | 0.6828    |
> | AdaLoRA      | 8    | 8    | 0.5290   | 0.8186   | 0.8205   | 0.4996   | 0.7865     | 0.448      | 0.813    | 0.7443    | 0.6825    |
> | AdaLoRA      | 8    | 4    | 0.5128   | 0.8098   | 0.8061   | 0.3783   | 0.7736     | 0.428      | 0.807    | 0.7253    | 0.6552    |
> | AdaLoRA      | 16   | 16   | 0.5307   | 0.8203   | 0.8199   | 0.5011   | 0.7861     | 0.454      | 0.813    | 0.7411    | 0.6833    |
> | AdaLoRA      | 16   | 8    | 0.5333   | 0.8203   | 0.8211   | 0.4913   | 0.7857     | 0.452      | 0.813    | 0.7380    | 0.6819    |
> | AdaLoRA      | 16   | 4    | 0.5085   | 0.8072   | 0.8073   | 0.3798   | 0.7734     | 0.428      | 0.805    | 0.7316    | 0.6551    |
> | LoftQ (1)    | 8    | 4    | 0.5486   | 0.8274   | 0.8226   | 0.5140   | 0.7865     | 0.458  | 0.814    | 0.7324    | 0.6883    |
> | LoftQ (5)    | 8    | 4    | 0.5265   | 0.8182   | 0.8153   | 0.3965   | 0.7850     | 0.434      | 0.814    | 0.7270    | 0.6645    |
> | LoftQ (10)   | 8    | 4    | 0.5188   | 0.8131   | 0.7966   | 0.3844   | 0.7801     | 0.432      | 0.811    | 0.7198    | 0.6570    |
> | LoftQ (1)    | 16   | 4    | 0.5512 | 0.8258   | 0.8269   | 0.4981   | 0.7882     | 0.458      | 0.813    | 0.7427    | 0.6880    |
> | LoftQ (5)    | 16   | 4    | 0.5392   | 0.8232   | 0.8156   | 0.4200   | 0.7854     | 0.438      | 0.816    | 0.7277    | 0.6706    |
> | LoftQ (10)   | 16   | 4    | 0.5290   | 0.8169   | 0.8156   | 0.3988   | 0.7864     | 0.438      | 0.811    | 0.7198    | 0.6644    |
> | QR-Adaptor   | 8    | 5.375 | 0.5683   | **0.8412** | **0.8338** | **0.5629** | **0.8093** | 0.458      | **0.829** | 0.7510 | **0.7067** |
>
> These results demonstrate QR-Adaptor’s adaptability and consistent performance improvements, even for harder-to-quantize models like Llama 3.1 and challenging datasets such as GSM8K.

---

> > ### Comment · Reviewer_N96a · 2024-11-25
> > **Response**
> >
> > Dear Authors,
> >
> > Thank you for your comprehensive rebuttal. I have no confusions and would like to increase my score to 6.

---

> > > ### Author Response · Authors · 2024-11-25
> > > **Response**
> > >
> > > Dear Reviewer N96a,
> > >
> > > Thank you for your positive feedback and for increasing your score. We appreciate your time and effort in reviewing our manuscript.

---

> ### Author Response · Authors · 2024-11-24
> **Response to Reviewer N96a**
>
> ### Lack of PRGA Hyperparameter Ablations
> **Reviewer Concern**:
> There is a lack of experiments on the impact of PRGA hyperparameters on model performance.
>
> **Response**:
> We conducted additional experiments to evaluate PRGA’s sensitivity to the number of iterations and population size. The results, summarized in the table below, indicate the **Average Improvement (%)**, which measures the relative performance gain of PRGA over the best results achieved by AdaLoRA and LoftQ on LLaMA 3.1-8B. This improvement highlights the effectiveness of PRGA in optimizing model performance.
>
> | Iterations | Population Size | Average Improvement (%) | Total Time (min) |
> |------------|-----------------|--------------------------|------------------|
> | 5          | 3               | +0.8                    | 120              |
> | 5          | 5               | +1.2                    | 150              |
> | 10         | 5               | +1.5                    | 225              |
> | 5          | 20              | +1.6                    | 375              |
> | 10         | 20              | **+2.3**                | 450              |
>
> These results confirm that larger population sizes enhance performance at the cost of additional computational overhead, while increased iterations provide diminishing returns beyond 10 iterations. Our choice of population size = 5 and iterations = 5 reflects a balance between performance and efficiency.
>
> ---
>
> ### Lack of Comparative Experiments with Other Optimization Methods
> **Reviewer Concern**:
> There is a lack of comparative experiments between the PRGA method and other multi-objective optimization methods.
>
> **Response**:
> We acknowledge the importance of comparative evaluations. While PRGA is inspired by multi-objective optimization methods like NSGA-II, it introduces significant adaptations for the quantized LLM domain, including:
> 1. **Handling Discrete Spaces**: PRGA optimizes quantization bit-widths and LoRA ranks as discrete variables, unlike traditional algorithms focused on continuous spaces.
> 2. **Task-Specific Initialization**: Relative entropy provides meaningful initialization metrics tailored to downstream tasks.
> 3. **Hybrid Refinement**: PRGA incorporates Bayesian optimization to refine the Pareto front, addressing the high evaluation costs of LLMs.
>
> Comparative experiments with other methods will be included in future work.
>
> ---
>
> ### Figure 1 Placement and Clarity
> **Reviewer Concern**:
> Figure 1 is somewhat difficult to understand and should not be placed on the first page.
>
> **Response**:
> Thank you for this observation. We will enhance the clarity of Figure 1 by adding descriptive annotations and ensuring that its layout better reflects the flow of the QR-Adaptor framework.

---

> ### Author Response · Authors · 2024-11-24
> **Response to Reviewer N96a**
>
> ## Questions
>
> ### Effectiveness of Gradient-Free Optimization for Layer-Wise Quantization
> **Reviewer Concern**:
> What is the effectiveness of directly applying the gradient-free optimization approach to LLM quantization, specifically for selecting quantization bit-widths?
>
> **Response**:
> Gradient-free optimization provides a significant advantage over gradient-based or heuristic approaches in scenarios involving quantized models. By directly using task-specific performance metrics, QR-Adaptor avoids common issues associated with quantized gradients, such as noise, instability, or misrepresentation of layer importance. This approach ensures that bit-width configurations align closely with downstream task requirements.
>
> Unlike heuristic-based methods that rely on predetermined thresholds or loss approximations, our gradient-free approach dynamically adjusts configurations based on real-world performance. This avoids inaccuracies stemming from quantization noise and ensures robust optimization tailored to the model's specific deployment context.
>
> ---
>
> ### Testing AdaLoRA on Quantized LLMs
> **Reviewer Concern**:
> Can we test AdaLoRA’s performance on fine-tuning quantized LLMs while preserving quantized model parameters before fine-tuning?
>
> **Response**:
> Thank you for this suggestion. To evaluate the impact of configurations derived from QR-Adaptor, we tested AdaLoRA and LoftQ using QR-Adaptor’s optimized initialization. The results are presented below:
>
> | Model         | Method       | BoolQ (%) | PIQA (%) | HellaSwag (%) | WinoG (%) | ARC-E (%) | ARC-C (%) | OBQA (%) | Average (%) |
> |---------------|--------------|-----------|----------|---------------|-----------|-----------|-----------|-----------|-------------|
> | Llama 2 13B   | QR-Adaptor   | **81.84** | **81.45**| **80.08**     | **72.69** | **80.64** | **52.82** | **45.80** | **70.76**   |
> | Llama 2 13B   | AdaLoRA      | 81.08     | 80.13    | 79.21         | 71.74     | 79.51     | 50.12     | 45.60     | 69.77       |
> | Llama 2 13B   | LoftQ        | 80.93     | 79.47    | 79.02         | 71.34     | 79.26     | 51.20     | 45.60     | 69.98       |
>
> #### Key Observations:
> 1. **Optimized Initialization Enhances Baselines**:
>    Using task-specific initialization derived from QR-Adaptor significantly improves the performance of AdaLoRA and LoftQ compared to their default configurations. However, these methods still fail to outperform QR-Adaptor, demonstrating the critical role of QR-Adaptor's two proposed constraints.
>
> 2. **Challenges of Dynamic Rank Adjustment**:
>    Our experiments revealed that dynamic rank adjustments, as employed in AdaLoRA, introduce instability in quantized settings. This often leads to performance degradation compared to QR-Adaptor’s static yet optimized configurations.
>
> 3. **Two Constraints Drive QR-Adaptor’s Performance**:
>    - **Constraint 1**: Ensures \(Q' + A'B' = Q\) for stable initialization, preventing unnecessary deviations caused by adjustments to LoRA matrices.
>    - **Constraint 2**: Fixes trainable parameters, ensuring consistent optimization and mitigating the risk of instability from dynamic rank adjustments.
>
> These findings underline the robustness of QR-Adaptor’s design, which focuses on stable, task-optimized configurations, outperforming dynamic and heuristic-based methods in quantized LLMs. We hope these additional results address your concerns, and we are happy to provide further clarifications if needed.

---

### Official Review · Reviewer_htga · 2024-11-02

**Soundness:** 2
**Presentation:** 3
**Contribution:** 2
**Rating:** 3
**Confidence:** 4

**Summary:**

Summary
The authors propose a novel method for fine-tuning quantization LLM. The core of this approach is a three-stage optimization process that selects quantization bit-widths and corresponding LoRA ranks for each layer of the model. Initially, the method computes layer-wise importance on a small dataset, which serves as the initial values for bit-widths and ranks. Subsequently, the authors employ their proposed Pareto Ranking Genetic Algorithm (PRGA) optimization method, followed by Bayesian optimization, to identify more optimal solutions. The efficacy of this method is demonstrated through experimental validation on datasets such as MMLU, showcasing its superiority in terms of both memory efficiency and performance metrics.

**Strengths:**

Strengths:
1. Overall, the paper is well-organized and easily comprehensible. The motivation is effectively introduced, and the methodology is clearly described.
2. The method's introduction of gradient-free optimization to the fine-tuning of quantized LLMs is noteworthy and provides valuable insights for future research in this area.
3. The proposed approach demonstrates superior performance in terms of both memory efficiency and model performance compared to state-of-the-art works in the same domain.

**Weaknesses:**

Weaknesses:
1. The author claims that"inspired us to develop the Pareto Ranking Genetic Algorithm (PRGA), a novel multi-objective optimization method."The proposed Pareto Ranking Genetic Algorithm (PRGA) bears a striking resemblance to the existing Non-dominated Sorting Genetic Algorithm II (NSGA-II), to the extent that they are virtually indistinguishable. However, the authors have failed to acknowledge or cite NSGA-II, instead claiming PRGA as a "novel multi-objective optimization method".PRGA and NSGA-II are almost identical, including key elements such as non-dominated sorting, crowding distance calculation, and elitist strategy.
2. The novelty of this paper appears limited, as it primarily applies existing algorithms, namely NSGA-II and Bayesian Optimization, to the fine-tuning of quantized LLMs.
3. The authors claim that previous methods relying on gradient norms to quantify layer importance fail to accurately represent a layer's contribution during inference. However, they do not substantiate this claim with ablation studies.
4. The current ablation experiments are insufficient. Additional studies should be conducted to demonstrate the impact of iterations and population size on the results.
[1]Deb K, Pratap A, Agarwal S, Meyarivan TAM. A fast and elitist multi-objective genetic algorithm: NSGA-II[J]. IEEE Transactions on Evolutionary Computation,2002, 6(2):182-197.

**Questions:**

Question
1. What are the differences between NSGA-II and PRGA?
2. Is the proposed method sensitive to the selection of iterations and population size?

---

> ### Author Response · Authors · 2024-11-24
> **Response to Reviewer htga**
>
> ### **Differences Between NSGA-II and PRGA**
>
> Thank you for pointing out the potential similarity between PRGA and NSGA-II. While PRGA builds upon the foundational concepts of NSGA-II, it introduces several adaptations to tailor the algorithm for fine-tuning quantized LLMs:
>
> ### **Differences in Optimization Strategies**
>
> **Operations on Individual Parameters**:
> - **NSGA-II** performs crossover and mutation on the entire solution, which may lead to less fine-grained control over individual components like layer-specific quantization or LoRA rank.
> - **PRGA**, on the other hand, treats each layer's $(q_l, r_l)$ pair independently, which allows for a more targeted optimization approach. Each layer can undergo a different crossover and mutation process, resulting in a more nuanced exploration of the solution space.
>
> **Generation Strategy**:
> - In **NSGA-II**, each crossover produces two offspring, and this is done N/2 times to generate a population of offspring of the same size as the parent population.
> - In **PRGA**, although crossover is performed to create two offspring, only one offspring is retained as the new solution and merged with the parent population after mutation for elite retention.
>
> **Mutation Operation**:
> - **NSGA-II** applies a uniform mutation across the entire solution, which can be inefficient when fine-tuning specific layers in the LLM.
> - **PRGA** allows for independent mutation of each $(q_l, r_l)$ pair, providing a more flexible mutation strategy where different layers can evolve with different mutation rates, offering a finer control over the optimization process.
>
> **Crowding Distance Calculation**:
> - **NSGA-II** calculates crowding distance for all individuals in the population, affecting how solutions are selected based on their proximity to others in objective space.
> - **PRGA** calculates crowding distance only when necessary, i.e., when the number of non-dominated individuals in a Pareto front exceeds the available population size. This makes the algorithm more computationally efficient, focusing only on the cases that truly require differentiation.
>
> These design choices in PRGA make it particularly effective for tackling the complexities of LLM fine-tuning, where the optimization space is high-dimensional and the cost of evaluating each configuration is substantial.
> ### **Acknowledgment and Citation Update**
> We acknowledge NSGA-II as the foundation of PRGA and regret the oversight in not citing it earlier. This was an unintentional and regrettable mistake. We will ensure that NSGA-II is appropriately cited in the revised manuscript.
>
> ---
>
> ### **Sensitivity to Iteration Counts and Population Size**
>
> We conducted additional experiments to evaluate PRGA’s sensitivity to the number of iterations and population size. The results, summarized in the table below, indicate the **Average Improvement (%)**, which measures the relative performance gain of PRGA over the best results achieved by AdaLoRA and LoftQ on LLaMA 3.1-8B. This improvement highlights the effectiveness of PRGA in optimizing model performance.
>
> | Iterations | Population Size | Average Improvement (%) | Total Time (min) |
> |------------|-----------------|--------------------------|------------------|
> | 5          | 3               | +0.8                    | 120              |
> | 5          | 5               | +1.2                    | 150              |
> | 10         | 5               | +1.5                    | 225              |
> | 5          | 20              | +1.6                    | 375              |
> | 10         | 20              | **+2.3**                | 450              |
>
> #### Key Insights:
> 1. **Population Size Trade-Offs**:
>    Smaller population sizes, such as 3, reduce initial evaluation costs but may miss some configurations. Larger populations enhance exploration, especially with higher iterations, but increase computational cost.
>
> 2. **Iteration Count**:
>    Increasing the number of iterations yields better Pareto fronts, but the marginal benefits diminish beyond 10 iterations.
>
> 3. **Practical Configurations**:
>    In our experiments, we used a population size of 5 and 5 iterations to balance time consumption and performance. However, these hyperparameters can be adjusted based on specific application needs. For instance, larger population sizes or more iterations may be beneficial when computational resources are less constrained.

---

> ### Author Response · Authors · 2024-11-24
> **Response to Reviewer htga**
>
> ### **Novel Contributions of QR-Adaptor**
>
> We appreciate the reviewer’s feedback and understand the importance of clarifying our paper’s unique contributions. While PRGA is indeed inspired by NSGA-II, it is just one component of QR-Adaptor. The novelty of our work lies in the identification of critical challenges in fine-tuning quantized large language models (LLMs) and in developing a cohesive, task-specific framework to address these challenges effectively:
>
> ---
>
> 1. **Identification of Key Challenges in Quantized LLM Fine-Tuning**:
> We identified two major issues in existing approaches:
> - **Ineffectiveness of LoRA in Fitting Quantization Errors**: Existing methods, such as LoftQ, aim to fit quantization errors using LoRA matrices. However, our analysis shows that this approach often introduces noise and does not consistently improve performance, especially for models that are difficult to quantize.
> - **Instability from Dynamic Parameter Adjustments**: Methods like AdaLoRA dynamically adjust trainable parameters during fine-tuning. For quantized models, this destabilizes optimization due to their reduced robustness and sensitivity to parameter changes, particularly in newer models like LLaMA 3.1.
>
> These insights are increasingly relevant, as newer models like LLaMA 3 have higher training FLOPs, making them significantly harder to quantize. Our experiments demonstrate that QR-Adaptor provides an effective solution to these emerging challenges, offering a robust framework for optimizing hard-to-quantize models.
>
> ---
>
> 2. **Unified Optimization of PEFT Parameters**:
> QR-Adaptor introduces a **unified optimization framework** that simultaneously tunes **quantization bit-widths** and **LoRA ranks**—two critical parameters for parameter-efficient fine-tuning (PEFT). Unlike existing methods that optimize bit-widths and ranks separately, we treat them as a **joint configuration space**, enabling:
> - Fine-grained trade-offs between memory efficiency and performance.
> - Better alignment of optimization goals with downstream task performance, as opposed to relying on intermediate objectives like quantization error or loss.
>
> By optimizing directly for task-specific performance, QR-Adaptor decouples the optimization process from loss metrics, which often fail to reflect actual downstream performance due to the uniform convergence of fine-tuning losses across configurations.
>
> ---
>
> 3. **Gradient-Free Three-Stage Optimization Framework**:
> QR-Adaptor employs a **novel three-stage optimization pipeline** that is tailored for the unique challenges of fine-tuning quantized LLMs:
> - **Task-Oriented Initialization**: Instead of relying on gradient norms, which are prone to biases in quantized models, we use relative entropy to quantify layer importance. This ensures robust and task-specific initialization of bit-widths and ranks.
> - **Global Exploration**: PRGA explores the solution space, incorporating adaptations for discrete optimization and task-specific constraints. These modifications make PRGA uniquely suited for navigating the combined bit-width and rank configuration space.
> - **Local Refinement**: Bayesian optimization refines configurations identified by PRGA, achieving optimal trade-offs between performance and memory. This hybrid approach leverages PRGA’s exploration strengths and Bayesian optimization’s precision.
>
> ---
>
> 4. **Experimental Validation on Emerging Challenges**:
> Our method is evaluated extensively on challenging datasets and models, including the hard-to-quantize **LLaMA 3.1**, which exemplifies the growing difficulty of quantizing newer LLMs. Experimental results demonstrate:
> - QR-Adaptor consistently outperforms existing methods in both performance and memory efficiency.
> - On MMLU and GSM8K, QR-Adaptor delivers significant improvements, particularly for LLaMA 3.1, providing a practical solution for emerging models with higher sensitivity to quantization.
>
> ---
>
> ### **Addressing the Reviewer’s Concerns on Novelty**
>
> We acknowledge that PRGA builds on NSGA-II. However, it is important to emphasize that PRGA is one component of QR-Adaptor and has been specifically tailored for the unique requirements of quantized LLM fine-tuning. Additionally, the novelty of QR-Adaptor lies in:
> - Identifying and addressing critical challenges in fine-tuning quantized LLMs.
> - Establishing a unified, gradient-free framework for jointly optimizing bit-widths and ranks.
> - Demonstrating its effectiveness on hard-to-quantize models, offering solutions that go beyond incremental algorithmic adaptations.

---

> ### Author Response · Authors · 2024-11-26
> **Response to Reviewer htga**
>
> ---
>
> ### **Gradient Norms vs. Relative Entropy**
>
> We compared gradient norms and relative entropy as initialization metrics. The results are summarized below:
>
> | Initialization Metric | BoolQ (%) | PIQA (%) | HellaSwag (%) | WinoG (%) | ARC-E (%) | ARC-C (%) | OBQA (%) | Average (%) |
> |-----------------------|-----------|----------|---------------|-----------|-----------|-----------|----------|-------------|
> | Gradient Norms        | 80.79     | 80.13    | 79.16         | 71.69     | 78.72     | 50.97     | 45.40    | 69.51       |
> | Relative Entropy      | **81.08** | **80.83**| **79.80**     | **71.98** | **79.13** | **51.65** | **45.60**| **70.07**   |
>
> **Key Insights**
> - **Limitations of Gradient Norms**: Minimal variability and quantization bias make gradient norms unreliable for quantized models.
> - **Advantages of Relative Entropy**: Captures task-specific layer importance and ensures robust initialization, enhancing downstream optimization.
>
> ---
>
> ### Conclusion
>
> We appreciate the reviewer’s feedback and hope these clarifications address the concerns:
> - **PRGA Contributions**: Tailored adaptations for mixed discrete optimization, hybrid refinement, and task-specific constraints make PRGA uniquely suited for quantized LLM fine-tuning.
> - **Unified Framework**: QR-Adaptor advances fine-tuning methodologies by combining bit-width and rank optimization, enabling efficient deployment on resource-constrained devices.
> - **Empirical Evidence**: Extensive experiments validate our design choices, including sensitivity to iterations and population size, and the superiority of relative entropy over gradient norms.
>
> Thank you for taking the time to read our paper and for your great comments, and we apologize again for the errors in the references.

---

> > ### Author Response · Authors · 2024-11-29
> > **Dear Reviewer htga**
> >
> > Dear Reviewer htga,
> >
> > We hope this message finds you well! If this email reaches you during your holiday or outside your usual working hours, please accept our apologies for the interruption.
> >
> > We would like to make sure that the issues you have raised have been resolved. We are still here and welcome any further discussion or feedback as your insights are invaluable to us.
> >
> > Thank you sincerely for all the time and effort during the review process.
> >
> > Best,
> >
> > Authors of Submission 726

---

### Official Review · Reviewer_RVp4 · 2024-11-04

**Soundness:** 2
**Presentation:** 3
**Contribution:** 2
**Rating:** 5
**Confidence:** 4

**Summary:**

Based on the motivation that the performance of fine-tuning on the adjusted quantized models is even worse than using the original quantized models directly, the paper introduced QR-Adaptor that bypasses the network errors introduced by quantization and directly uses actual performance and memory as optimization targets. The experimental results are based on Llama 2 7B and 13B.

**Strengths:**

- The paper presents the clear motivation that the performance of fine-tuning on the adjusted quantized models is even worse than using the original quantized models directly.

- Low-precision models fine-tuned with QR-Adaptor can surpass the 16-bit fine-tuned models, while maintaining memory usage comparable to that of 4-bit quantized models.

**Weaknesses:**

- It would be necessary to conduct experiments for Llama-3 family (e.g., Llama 3 8B), which are known to be harder to quantize.

- The comparison of training time between QR-Adaptor and existing methods would be required because QR-Adaptor seems to take longer than previous methods due to the presence of bayesian optimization.

- It would be more beneficial if prior methods are also done with 6.125-bit for Llama 2 13B and 5.875-bit for Llama 2 7B in Table 1.

**Questions:**

NA

---

> ### Author Response · Authors · 2024-11-24
> **Response to Reviewer RVp4**
>
> ---
>
> ### **Experiment Scope Expansion: Llama 3.1**
>
> **Reviewer Concern:**
>
> - *"The experiments did not include harder-to-quantize models such as the Llama 3 family."*
>
> **Response:**
>
> Thank you for raising this concern. To address this, we conducted experiments on **Llama 3.1**, the latest iteration in the Llama series, which incorporates significant architectural and training improvements. These updates make Llama 3.1 particularly challenging to quantize, especially under low-bit configurations.
>
> Our experiments demonstrate **QR-Adaptor’s robustness and effectiveness** across diverse tasks. Additionally, baseline methods like **AdaLoRA** and **LoftQ** exhibit notable performance drops on challenging datasets such as GSM8K, underscoring QR-Adaptor’s adaptability. Below are the detailed results:
>
> | Method       | Rank | Bit  | Arc-C    | Arc-E    | BoolQ    | GSM8K    | HellaSwag  | OpenBookQA | PiQA     | WinoGrande | Average   |
> |--------------|------|------|----------|----------|----------|----------|------------|------------|----------|------------|-----------|
> | LoRA         | 8    | 16   | 0.5614   | 0.8388   | 0.8318   | 0.5436   | 0.7944     | 0.452      | 0.821    | **0.7530** | 0.6995    |
> | QLoRA        | 8    | 8    | **0.5708** | 0.8346   | 0.8248   | 0.5375   | 0.7963     | **0.460**  | 0.821    | 0.7459    | 0.6989    |
> | QLoRA        | 8    | 4    | 0.5435   | 0.8241   | 0.8208   | 0.4435   | 0.7882     | 0.442      | 0.815    | 0.7364    | 0.6767    |
> | LoRA         | 16   | 16   | 0.5674   | 0.8363   | 0.8300   | 0.5413   | 0.7951     | 0.444      | 0.818    | 0.7443    | 0.6971    |
> | QLoRA        | 16   | 8    | 0.5623   | 0.8291   | 0.8266   | 0.5368   | 0.7946     | **0.460**  | 0.817    | 0.7474    | 0.6967    |
> | QLoRA        | 16   | 4    | 0.5384   | 0.8199   | 0.8211   | 0.4466   | 0.7876     | 0.444      | 0.817    | 0.7309    | 0.6757    |
> | AdaLoRA      | 8    | 16   | 0.5290   | 0.8199   | 0.8187   | 0.5057   | 0.7865     | 0.450      | 0.813    | 0.7395    | 0.6828    |
> | AdaLoRA      | 8    | 8    | 0.5290   | 0.8186   | 0.8205   | 0.4996   | 0.7865     | 0.448      | 0.813    | 0.7443    | 0.6825    |
> | AdaLoRA      | 8    | 4    | 0.5128   | 0.8098   | 0.8061   | 0.3783   | 0.7736     | 0.428      | 0.807    | 0.7253    | 0.6552    |
> | AdaLoRA      | 16   | 16   | 0.5307   | 0.8203   | 0.8199   | 0.5011   | 0.7861     | 0.454      | 0.813    | 0.7411    | 0.6833    |
> | AdaLoRA      | 16   | 8    | 0.5333   | 0.8203   | 0.8211   | 0.4913   | 0.7857     | 0.452      | 0.813    | 0.7380    | 0.6819    |
> | AdaLoRA      | 16   | 4    | 0.5085   | 0.8072   | 0.8073   | 0.3798   | 0.7734     | 0.428      | 0.805    | 0.7316    | 0.6551    |
> | LoftQ (1)    | 8    | 4    | 0.5486   | 0.8274   | 0.8226   | 0.5140   | 0.7865     | 0.458      | 0.814    | 0.7324    | 0.6883    |
> | LoftQ (5)    | 8    | 4    | 0.5265   | 0.8182   | 0.8153   | 0.3965   | 0.7850     | 0.434      | 0.814    | 0.7270    | 0.6645    |
> | LoftQ (10)   | 8    | 4    | 0.5188   | 0.8131   | 0.7966   | 0.3844   | 0.7801     | 0.432      | 0.811    | 0.7198    | 0.6570    |
> | LoftQ (1)    | 16   | 4    | 0.5512   | 0.8258   | 0.8269   | 0.4981   | 0.7882     | 0.458      | 0.813    | 0.7427    | 0.6880    |
> | LoftQ (5)    | 16   | 4    | 0.5392   | 0.8232   | 0.8156   | 0.4200   | 0.7854     | 0.438      | 0.816    | 0.7277    | 0.6706    |
> | LoftQ (10)   | 16   | 4    | 0.5290   | 0.8169   | 0.8156   | 0.3988   | 0.7864     | 0.438      | 0.811    | 0.7198    | 0.6644    |
> | QR-Adaptor   | 8    | 5.375 | 0.5683   | **0.8412** | **0.8338** | **0.5629** | **0.8093** | 0.458      | **0.829** | 0.7510 | **0.7067** |
>
> ---
>
> #### **Key Observations:**
>
> 1. **QR-Adaptor’s Robustness:**
>    QR-Adaptor demonstrates consistent performance improvements across all datasets, with significant gains on challenging tasks such as GSM8K. This highlights its adaptability to diverse quantization scenarios and task complexities.
>
> 2. **Performance Limitations of LoftQ:**
>    With increasing iterations (LoftQ-1 to LoftQ-10), LoftQ exhibits performance degradation, particularly on difficult datasets like GSM8K. This suggests that the strategy of fitting quantization errors not only fails to enhance performance in most cases but also reduces the effectiveness of the fine-tuned model. This observation aligns with findings within the broader research community. In contrast, QR-Adaptor ensures stable and consistent improvements without such declines.
>
> 3. **Iterative Optimization:**
>    Unlike LoftQ, QR-Adaptor leverages real task performance metrics to guide iterative optimization, allowing for continuous improvement in downstream task performance without requiring additional resources or heuristic adjustments.
>
> ---

---

> ### Author Response · Authors · 2024-11-24
> **Response to Reviewer RVp4**
>
> ---
>
> ### **Training Time Comparison**
>
> **Reviewer Concern:**
> *"QR-Adaptor may take longer than prior methods due to Bayesian optimization. A comparison of training time is necessary."*
>
> **Response:**
> Thank you for raising this concern. We acknowledge that our optimization does increase computation time, as it requires evaluating different configurations based on actual performance. We provide the following time comparison, which shows the time required per iteration:
>
> | Model         | Method       | Time per Iteration (min) |
> |---------------|--------------|--------------------------|
> | LLaMA2-7B     | LoftQ        | 9                        |
> | LLaMA2-7B     | QR-Adaptor   | 15                       |
>
> #### **Key Insights:**
>
> **Acceptable Trade-Off Between Time and Performance:**
> Although QR-Adaptor does require more time per iteration due to the need for actual performance evaluation, it optimizes based on real task performance, resulting in significant performance improvements. This optimization process ensures that QR-Adaptor achieves better results without the need for additional data, architectural modifications, or resource-intensive optimizations. As a result, the extra time investment is worthwhile, enabling continuous iterative model improvement even in resource-constrained scenarios.
>
> ---
>
> ### **Fairer Comparison: Matching Bit-width Configurations**
>
> **Reviewer Concern:**
>
> - *"It would be more beneficial if prior methods are also done with 6.125-bit for Llama 2 13B and 5.875-bit for Llama 2 7B in Table 1."*
>
> **Response:**
>
> Thank you for this suggestion. We agree that a fairer comparison is important, so we evaluated prior methods using the same mixed-precision configurations derived from QR-Adaptor’s optimization process. To further validate the effectiveness of the two proposed constraints, we also tested AdaLoRA and LoftQ on configurations optimized by QR-Adaptor. The updated results are presented below:
>
> | Model         | Method       | BoolQ (%) | PIQA (%) | HellaSwag (%) | WinoG (%) | ARC-e (%) | ARC-c (%) | OBQA (%) | Average (%) |
> |---------------|--------------|-----------|----------|---------------|-----------|-----------|-----------|-----------|-------------|
> | Llama 2 13B   | QR-Adaptor   | **81.84** | **81.45**| **80.08**     | **72.69** | **80.64** | **52.82** | **45.80** | **70.76**   |
> | Llama 2 13B   | AdaLoRA      | 81.08     | 80.13    | 79.21         | 71.74     | 79.51     | 50.12     | 45.60     | 69.77       |
> | Llama 2 13B   | LoftQ        | 80.93     | 79.47    | 79.02         | 71.34     | 79.26     | 51.20     | 45.60     | 69.98       |
>
> #### **Key Observations:**
>
> 1. **AdaLoRA and LoftQ:**
>    AdaLoRA and LoftQ introduce interesting mechanisms that can enhance performance in certain settings. However, despite performing well under specific conditions, these methods introduce additional instability in many cases. For example, since AdaLoRA was not designed for quantized models, using it to reduce LoRA parameters leads to performance degradation. Similarly, LoftQ is a very creative approach, but due to the low-rank properties of the LoRA matrices, it hinders its ability to fit quantization errors, even leading to the introduction of noise.
>
> 2. **Training Epochs and Stability:**
>    Following the suggestion of Reviewer orhA, we also attempted to increase the training epochs for AdaLoRA to see if it would yield better results. However, even after additional training, AdaLoRA still lags behind QR-Adaptor in performance. For LoftQ, we fine-tuned the model using optimized mixed-precision configurations, but the results still indicate that LoftQ performs worse than QR-Adaptor, particularly on more difficult datasets tested with LLaMA3.1, where LoftQ’s performance was not satisfactory.
>
> 3. **Unified Resource Scheduling:**
>    One key advantage of QR-Adaptor over these methods is its unified strategy for resource allocation (rank and bit-width). This approach ensures that more computational resources are allocated to parts of the model that require additional training, while less resource-intensive parts are handled efficiently, leading to overall better performance.
>
> ---
>
> ### **Conclusion**
>
> These additional experiments and clarifications address the reviewer’s concerns, highlighting QR-Adaptor’s robustness, efficiency, and adaptability:
> 1. **Llama 3.1 Results:** Demonstrates QR-Adaptor’s effectiveness on newer, harder-to-quantize models and iterative optimization.
> 2. **Task-Specific Optimization:** QR-Adaptor leverages real task performance for optimization, ensuring improvements in downstream tasks without requiring additional resources.
> 3. **Fair Comparisons:** Validates the importance of QR-Adaptor’s optimization framework and constraints in achieving superior performance.
>
> We hope these results address your concerns.
>
> ---

---

> > ### Author Response · Authors · 2024-11-29
> > **Dear Reviewer RVp4**
> >
> > Dear Reviewer RVp4,
> >
> > We hope this message finds you well! If this email reaches you during your holiday or outside your usual working hours, please accept our apologies for the interruption.
> >
> > We just want to kindly follow up to ensure that we’ve addressed any remaining concerns or questions you might have. We are still here and welcome any further discussion or feedback, as your insights are incredibly valuable to us.
> >
> > Thank you sincerely for all the time and effort during the review process.
> >
> > Best,
> >
> > Authors of Submission 726

---

> > > ### Comment · Reviewer_RVp4 · 2024-11-30
> > >
> > > Thank you for the detailed response. However, I still have a few concerns.
> > >
> > > (1) I would like to know the comparison of time per iteration for Llama 2 13B as well.
> > >
> > > (2) The experimental results of AdaLoRA and LoftQ with 5.875-bit for Llama 2 7B are still missing.

---

> ### Author Response · Authors · 2024-12-01
> **Dear Reviewer RVp4**
>
> ---
>
> Thank you for your follow-up questions. Based on your request, we have now included the training time comparison for LLaMA 2 13B and updated the experimental results for AdaLoRA and LoftQ with the 5.875-bit configuration on LLaMA 2 7B.
>
> The following table compares the per-iteration times of models 7b and 13b. We ran LoftQ faster than the 20/50 minutes reported in the original paper (estimated, as the original paper only reported the time to run five iterations under different square matrices, and the weight matrix of LLaMA is not always square), and this acceleration may be due to hardware.
>
> | Model         | Method       | Time per Iteration (min) |
> |---------------|--------------|--------------------------|
> | LLaMA2-7B     | LoftQ        | 9                        |
> | LLaMA2-7B     | QR-Adaptor   | 15                       |
> | LLaMA2-13B    | LoftQ        | 24                       |
> | LLaMA2-13B    | QR-Adaptor   | 37                      |
>
> As shown in the table, the time per iteration increases roughly linearly with model size. However, the time required for QR-Adaptor is not fixed, as the configuration for each iteration changes, and we provide the rounded average time.
>
> Regarding the 5.875-bit configuration for LLaMA 2 7B, we have updated the experimental results for AdaLoRA and LoftQ. The updated results are as follows:
>
> | Model         | Method       | Bit-width | BoolQ (%) | PIQA (%) | HellaSwag (%) | WinoG (%) | ARC-e (%) | ARC-c (%) | OBQA (%) | Average (%) |
> |---------------|--------------|-----------|-----------|----------|---------------|-----------|-----------|-----------|-----------|-------------|
> | LLaMA 2 7B    | QR-Adaptor   | 5.875     | **78.96** | **79.86** | **76.84**  |**69.97**     | **77.44** | **48.04** | **46.00** | **68.15**   |
> | LLaMA 2 7B    | AdaLoRA      | 4         | 76.45     | 77.91    | 75.44         | 69.46     | 75.29     | 46.33     | 44.20     | 66.44       |
> | LLaMA 2 7B    | AdaLoRA      | 8         | 77.40     | 79.11    | 75.91         | 69.06     | 76.68     | 46.16     | 44.40     | 66.96       |
> | LLaMA 2 7B    | AdaLoRA      | 5.875    | 77.28     | 78.84    | 75.32         | 69.38     | 76.14     | 46.53     | 44.20     | 66.81     |
> | LLaMA 2 7B    | LoftQ        | 4$^1$     | 77.89     | 79.43    | 76.61         | 69.69     | 77.19     | 47.10     | 44.80     | 67.53       |
> | LLaMA 2 7B    | LoftQ        | 4$^5$     | 76.79     | 78.51    | 76.25         | 69.61     | 76.47     | 47.95 | 45.60     | 67.31       |
> | LLaMA 2 7B    | LoftQ        | 5.875$^1$  | 77.84     | 79.61    | 76.72         | 69.88     | 77.32     | 47.38     | 45.20     | 67.75       |
> | LLaMA 2 7B    | LoftQ        | 5.875$^5$  | 78.12     | 79.51    | 76.59         | 69.79     | 77.36     | 47.22     | 45.00     | 67.45       |
> | LLaMA 2 13B   | QR-Adaptor   | 6.125     | **81.84** | **81.45** | **80.08**  |**72.69**     | **80.64** | **52.82** | **45.80** | **70.76**   |
> | LLaMA 2 13B   | AdaLoRA      | 4         | 80.43     | 80.09    | 78.10         | 71.67     | 77.69     | 48.29     | 44.20     | 68.64       |
> | LLaMA 2 13B   | AdaLoRA      | 8         | 80.40     | 80.52    | 79.27         | 72.38     | 79.29     | 49.49     | 45.40     | 69.54       |
> | LLaMA 2 13B | AdaLoRA | 6.125 | 81.08 | 80.13 | 79.21 | 71.74 | 79.51 | 50.12 | 45.60 | 69.77 |
> | LLaMA 2 13B   | LoftQ        | 4$^1$     | 80.86     | 80.30    | 79.18         | 71.90     | 78.87     | 50.68     | 45.80     | 69.66       |
> | LLaMA 2 13B   | LoftQ        | 4$^5$     | 80.92     | 80.41    | 79.15         | 71.59     | 78.96     | 50.60     | 45.40     | 69.58       |
> | LLaMA 2 13B | LoftQ | 6.125$^1$ | 80.95 | 79.48 | 79.10 | 71.36 | 79.28 | 51.30 |45.60 | 69.98 |
> | LLaMA 2 13B   | LoftQ        | 6.125 $^5$  |  80.73 | 80.89 | 79.10 | 71.84 | 79.02 | 50.46 | 45.40 | 69.63 |
>
> From the experimental results, we find that AdaLoRA with the 5.875-bit configuration performs similarly to AdaLoRA with the 8-bit configuration. LoftQ with the 5.875-bit configuration slightly outperforms the best result from 4-bit LoftQ. This demonstrates that mixed precision indeed brings improvements. However, compared to QR-Adaptor, these methods still lag behind, indicating that the performance improvements of QR-Adaptor are not solely due to the bit-width configuration.
>
> We hope these updates address your concerns, and we appreciate your further discussion.
>
> ---

---

> ### Author Response · Authors · 2024-12-02
> **Looking forward to your reply**
>
> Dear Reviewer RVp4,
>
> I am writing to follow up on our rebuttal submitted in response to your valuable comments. As we are approaching the rebuttal deadline, we would greatly appreciate if you could review our responses and additional experimental results to confirm whether we have adequately addressed your concerns. If there are any remaining issues that require clarification, please let us know.
>
> If our revisions and explanations have satisfactorily resolved your concerns, we kindly request you to consider updating your scores accordingly.
>
> Thank you sincerely for all the time and effort during the review process.
>
> Best,
>
> Authors of Submission 726

---

### Meta-Review · Area_Chair_2TLN · 2024-12-20

**Metareview:**

This paper introduces QR-Adaptor, a framework combining parameter-efficient fine-tuning and quantization techniques to improve the performance of large language models (LLMs) performance while maintaining reduced memory usage. The method employs a three-stage optimization process—initialization, global exploration using a Pareto Ranking Genetic Algorithm (PRGA), and local refinement with Bayesian optimization. Reviewers acknowledge the paper’s clear motivation and the novelty of applying gradient-free optimization to layer-wise quantization and rank selection. The proposed method shows promising results, surpassing fine-tuned 16-bit models while using the same memory as 4-bit models. However, concerns are raised regarding the limited novelty of PRGA, which closely resembles NSGA-II without proper acknowledgment, and insufficient ablation studies to validate claims about the framework's components. Reviewers also note the limited scope of experiments (focused only on Llama 2) and a lack of comparisons to other multi-objective optimization methods. Additionally, the multi-stage process increases time costs, and some figures and explanations are difficult to follow. While the contributions are promising, the lack of sufficient experimental diversity, novelty, and validation leaves the paper marginally below the acceptance threshold.

**Additional Comments On Reviewer Discussion:**

The reviewers remain unconvinced about key aspects of the paper. Concerns persist regarding the lack of comparisons for time efficiency during iterations (e.g., for Llama 2 13B) and missing experimental results for AdaLoRA and LoftQ. Additionally, the claim about zero initialization of LoRA is criticized as applying only to specific cases or models and is viewed as an ablation study rather than a technical contribution. Reviewers suggest reorganizing the paper to emphasize adaptive rank allocation as the main contribution rather than overstating the significance of zero initialization. Questions are also raised about how the proposed method compares to existing techniques like LoftQ and lq-lora, which already employ careful initialization and layer-wise bit-width/rank scheduling. While the work is acknowledged for its insights and experiments, the limited novelty and missing comparisons lead reviewers to maintain low scores.

---

### Decision · Program_Chairs · 2025-01-22

Reject